# Captivated by thought: "Sticky" thinking leaves traces of perceptual decoupling in task-evoked pupil size

**Stefan Huijser** *, Mathanja Verkaik, Marieke K. van Vugt, Niels A. Taatgen

Bernoulli Institute for Mathematics, Computer Science, and Artificial Intelligence, University of Groningen, Groningen, Netherlands

* stefanhuijser@outlook.com

**Data Availability Statement:** The data, materials, and analysis code are publicly available online at the Open Science Framework (link to project: https://osf.io/m6ujg/).

## Abstract

Throughout the day, we may sometimes catch ourselves in patterns of thought that we experience as rigid and difficult to disengage from. Such "sticky" thinking can be highly disruptive to ongoing tasks, and when it turns into rumination constitutes a vulnerability for mental disorders such as depression and anxiety. The main goal of the present study was to explore the stickiness dimension of thought, by investigating how stickiness is reflected in task performance and pupil size. To measure spontaneous thought processes, we asked participants to perform a sustained attention to response task (SART), in which we embedded the participant's concerns to potentially increase the probability of observing sticky thinking. The results indicated that sticky thinking was most frequently experienced when participants were disengaged from the task. Such episodes of sticky thought could be discriminated from neutral and non-sticky thought by an increase in errors on infrequent no-go trials. Furthermore, we found that sticky thought was associated with smaller pupil responses during correct responding. These results demonstrate that participants can report on the stickiness of their thought, and that stickiness can be investigated using pupillometry. In addition, the results suggest that sticky thought may limit attention and exertion of cognitive control to the task.

## Introduction

### Background

In response to pressing concerns and unreached goals we may catch ourselves in thoughts that we feel are difficult to disengage from. For example, we may be absorbed in thinking about a recently received paper rejection, while we should actually be reading this article. In general, task-unrelated thought is referred to as *mind wandering* [1]. However, in cases such as the paper rejection, these thoughts may not leave us alone, and make it very difficult to concentrate on our immediate tasks. In this case, one can call these thoughts *sticky* [2,3]. An extreme form of such sticky thought is rumination, a rigid and narrow-focused thought process that is hard to disengage from and often negative in valence and self-related [4]. In general,

**Funding:** This research was supported by a grant from the European Research Council (MULTITASK - 283597; https://erc.europa.eu/) awarded to N.A. Taatgen. The funders had no role in study design, data collection and analysis, decision to publish, or preparation of the manuscript.

**Competing interests:** The authors have declared that no competing interests exist.

rumination causes individuals to be unable to concentrate and devote their attention to tasks at hand because attention is focused internally instead [2]. However, in contrast to depressive rumination, sticky thoughts could also have a positive valence, for example when we are caught up in a pleasant fantasy that we do not want to let go of, or thoughts with desire for a delicious cookie keep recurring in our minds [5,6]. Another term for sticky thought is perseverative cognition. Perseverative cognition has been associated with activation of the physiological stress system, and has been proposed to play a key role in the onset and maintenance of depression [7] and anxiety [8,9]. Finally, sticky thought is closely related to the concept of *constrained thinking* [10,11]. Constrained thinking refers to an experience in which thoughts do not move freely but instead are focused on a narrow set of content. It is different from our concept of sticky thinking in the question that is posed to the participant—while sticky refers to the experience of the participant that it is difficult to drop the current stream of thought, constrained refers to participants' experience of having a stream of thought that is–deliberately or not– restricted to a narrow set of content.

Yet, sticky thoughts—especially in their non-clinical form—could also have advantages to the individual. By temporarily shielding thought from external distractions, they can help the individual to work on future goals [12–15]. When goals remain unattained and concerns unresolved but the thoughts remain, sticky thoughts may become increasingly more intrusive, disrupting our everyday functioning [2,3,16]. Therefore, sticky thoughts may have important effects on our performance in everyday tasks.

Sticky thought has mostly received attention in literature on psychopathology. Studies have demonstrated that perseverative cognition has measurable negative effects on somatic health (for a review see, [17]). For example, rumination and worry are associated with prolonged activation of the immune system [18], decreases in heart rate variability [19], and increases in blood pressure [20]. Hence, sticky thoughts may not only be disruptive to task performance, but also pose a risk for developing mental and somatic health issues.

## Examining stickiness of thought with self-report and task performance

Despite the known disruptive effects of sticky thought on task performance, we have limited understanding of the (attentional) processes that are associated with sticky thought and how those differ from non-sticky thought. One reason for this is that sticky thought is challenging to detect in the context of an experiment [1]. Sticky thinking is largely a covert process that leaves few directly observable signs. Indeed, related processes such as perseverative cognition and rumination have mostly been investigated using self-report questionnaires that measure trait rumination or worry (i.e., the general tendency to engage in sticky thinking), or alternatively, by asking the participant to report on the frequency of ruminative or worry episodes retrospectively. Correlating such measures with task performance [3,21,22] and neurocognitive measures (e.g., [19]) has yielded valuable insights (see [7]). For example, Beckwé et al. [22] found that in an exogenous cue task (ECT) participants with a strong tendency to ruminate had longer reaction times following invalid negative personality trait cues, suggesting that such participants experience more difficulty to disengage from negative personality trait cues, likely because these cues set off a train of negative self-related thinking. Aside from this cognitive inflexibility, studies with cardiac measures have shown that rumination is associated autonomic rigidity, demonstrated by persistent low heart rate variability [19,20]. Despite these insights from questionnaire-based measures of sticky thoughts, self-report arguably lacks precision, given limitations in memory that bias reporting [23,24] and participants' tendency to produce socially desirable answers. Furthermore, because questionnaires only provide a single after-the-fact measure, it is not possible to compare sticky with non-sticky thought within an individual.

A different, and potentially better, method to measure sticky thought are thought probes. Thought probes are short self-report questionnaires that are embedded in a task to measure the content and dynamics of current thought at various points in time during an on-going task [25,26]. They have the advantage that experiences can be caught close to when they arise. Furthermore, they allow for repeated measures of experienced thought making it possible to investigate changes in thought content over the course of the experiment. For example, Unsworth and Robison [27] used thought probes to investigate how different attentional states, such as mind wandering and external distraction, correlated with task performance and pupil size measures in sustained attention task. The researchers observed that task performance decreased and pupil size became smaller with time-on-task. Also, they found that reports of mind wandering were more frequent when the experiment progressed. This demonstrates that time-on-task influences are important to consider when studying self-generated thinking.

So far, we are familiar with only one study that used thought probes to investigate sticky thought. Van Vugt and Broers [2] used thought probe responses in conjunction with task performance measures to investigate how self-reported stickiness of thought was associated with the probability of being disengaged from the task (i.e., off-task). In addition, the researchers examined how self-generated thought and its stickiness affected performance. They asked participants to perform a variation of a go/no-go task referred to as the sustained attention to response task (SART; [28]). This task is suitable for studying self-generated thought because it is slow-paced and induces habitual responding, therefore allowing self-generated thought to occur. In line with their expectation, self-reported stickiness of thought increased the probability of being disengaged from the task and negatively influenced performance. Stickiness of thought was associated with more variable response times. Previous research has also indicated that variability in response times may be a relevant correlate for self-generated thought (see e.g., [29–31]). The increase in variability may indicate that participants allocate less attention to the task, resulting in reactive and more variable responding [32]. All in all, this study demonstrated that stickiness of thought is a relevant dimension of self-generated thought. Furthermore, it indicated that people can meaningfully report on the stickiness of their thought.

## Correlates of self-generated thought and stickiness in pupillometry

In addition to task performance, neurocognitive measures can be used to detect sticky thought, and can provide insight into the processes and mechanisms associated with sticky thought. In this study, we will use pupillometry to gain insight in sticky thought. Pupil size is an interesting measure because it relatively unobtrusive and easy to record. Furthermore, research has indicated that lapses of attention can be distinguished in various pupil size measures [33,34]. Therefore, we may also be able to detect differences in pupil size depending on the stickiness of thought.

Pupil size is typically measured on two temporal scales, reflecting different cognitive or neural processes. The most common of these measures is the task-evoked response in pupil size. The task-evoked response is a transient increase in pupil size following the processing of a task event, peaking at around 1s after event onset [35]. The magnitude of this response has been demonstrated to depend on the amount of attention, cognitive control, and cognitive processing required by the task [36–38]. Research has consistently found that when we engage in self-generated thought, our evoked responses in pupil size are smaller [33,39–41]. Smallwood and colleagues [15] interpreted this smaller response in pupil size as evidence that external processing is being inhibited during self-generated thinking, so-called *perceptual decoupling*.

In addition to stimulus-evoked pupil responses, pupil size is also measured during task-free periods, referred to as baseline or tonic pupil size. Baseline pupil size is proposed to reflect

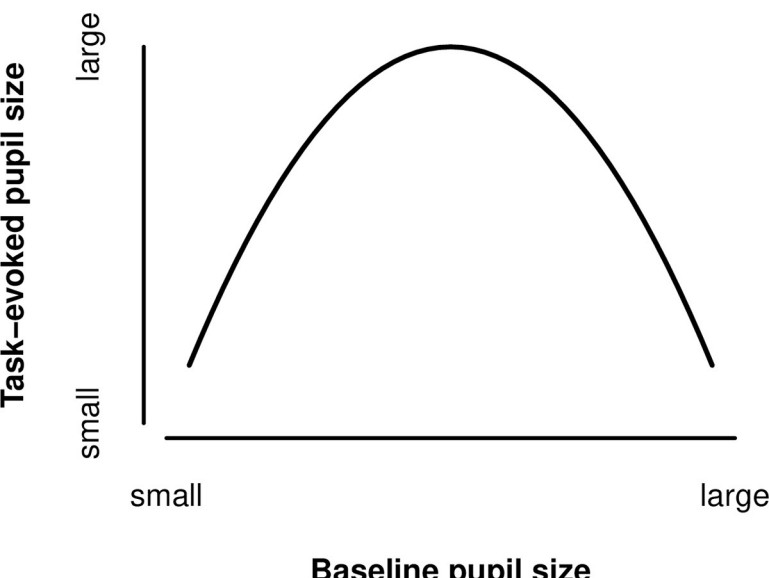

**Baseline pupil size**

**Fig 1. Adaptive gain curve.** The adaptive gain curve describes the relationship between baseline pupil size and task-evoked pupil size [43,44]. Task-evoked responses are maximized at intermediate levels of baseline pupil size, but decrease in magnitude when the baseline is smaller or larger. The curve also makes predictions about task performance. Performance on a task is optimal at intermediate baseline pupil size when task-evoked responses are maximal. Task performance decreases when baseline pupil size is smaller or larger than the intermediate level.

locus coeruleus norepinephrine (LC-NE) system functioning, which has been associated with controlling overall arousal levels and the tendency to seek for novelty [27, see 42,43]. Large baseline pupil size has been correlated with high tonic LC-NE firing, indicating a state of over-arousal and tendency to explore new behaviors. On the other hand, smaller baseline pupil sizes have been related to low tonic firing, under-arousal, and inactivity. Interestingly, research has proposed that the relationship between baseline pupil size, task-evoked pupil size, and task performance can be described with an adaptive gain curve (see Fig 1; see e.g., [37,44]). Task performance is optimal at intermediate levels of baseline pupil size when task-evoked responses are maximal. Task performance decreases when baseline pupil size is either larger or smaller.

Since stickiness (i.e., the difficulty in disengaging from thought) is a novel topic, no studies have directly investigated how stickiness is reflected in baseline and task-evoked pupil size. Nonetheless, predictions can be made based on related research. Given the disruptiveness of sticky thought to ongoing activities, we may expect that sticky thought, similar to self-generated thinking, is associated with smaller task-evoked responses in pupil size. As predicted by adaptive gain (see Fig 1 above), a smaller task-evoked response in pupil size with episodes of sticky thought would imply that the thought process is associated with either smaller or larger than average baseline pupil size. However, which one is open to debate. In clinical samples, Siegle et al. [45] found that rumination was associated with larger baseline pupil sizes. The researchers hypothesized that this larger baseline pupil size reflected sustained emotional processing [46]. In contrast, Konishi et al. [47] found that in non-clinical samples, negative and intrusive thoughts were associated with smaller baseline pupil size [48]. One recent study investigated how the "intensity" of experienced thought was reflected in baseline pupil size, which may be a dimension somewhat comparable to the stickiness of the thought [41]. However, that study was unsuccessful in finding an effect of intensity of thought in baseline pupil size. Also for self-generated thought, the literature has not reached consensus on where the

thought process lies on the adaptive curve [15,40,49]. Given that sticky thought is proposed to develop from thinking about pressing concerns and unreached goals [13,16], one might think that sticky thought is associated with high arousal, and therefore larger than average baseline pupil size. On the other hand, one might also think that sticky thought results from a state of inertia, reflected in smaller baseline pupil size.

### The current study

The main goal of the present study was to investigate how stickiness of thought is reflected in task performance and pupillary measures (i.e., baseline and task-evoked response in pupil size). We asked participants to perform a variation of the SART, in which we embedded the personal concerns of participants in the task to potentially increase the tendency for sticky self-generated thought [see 50]. We included periodic thought probes in the SART to measure what participants were currently thinking about (i.e., *attentional state*) and how difficult it was to disengage from the thought (i.e., *stickiness* of thought).

In line with previous work, we expected that sticky thought would be associated with being more disengaged from the task. Since being disengaged from the task has been found to reduce no-go accuracy, speed-up response times and increase RTCV, we predicted that no-go accuracy would decrease, response times would be faster, and that RTCV would increase with reported stickiness of thought.

With respect to pupillary measures, we predicted that stickiness of thought would be associated with smaller task-evoked responses in pupil size, indicating reduced attention to the SART. Given that no research has investigated the influence of stickiness of thought on baseline pupil size, and the inconsistency in previous studies that tried to relate for baseline pupil size to self-generated thought, we formulated no prior hypotheses for that measure.

## Materials and methods

### Participants

We recruited 34 Native Dutch speakers for this experiment (20 female; *M age* = 22.7, *SD age* = 2.7). Participants were recruited from a paid research participant pool on Facebook, as well as from the Artificial Intelligence Bachelor and Master programs at the University of Groningen. We screened the participants for having normal or corrected-to-normal vision prior to testing. All participants provided informed consent at the start of the laboratory session. The experiment was conducted in accordance with the Declaration of Helsinki and approved by the Ethical Committee of Psychology (ECP) at the University of Groningen (research code: pop-015-170). Written informed consent was obtained prior to the experiment for each participant.

### Materials

**Questionnaire session.**  Participants were requested to fill out three online questionnaires prior to the experiment. Since we wanted to maximize the probability of observing sticky thinking, which is known to often be related to concerns and worries, we adopted the current concerns manipulation by McVay & Kane [50]. For this manipulation, individual current concerns were collected using an online version of the Personal Concerns Inventory (PCI; adapted from [51]). In this questionnaire, participants were asked to write down short statements about current goals or concerns in eight different areas, including: 1) home and household matters, 2) employment and finances, 3) partner, family, and relatives, 3) friends and acquaintances, 4) spiritual matters, 5) personality matters, 6) education and training, 7) health and medical matters, and 8) hobbies, leisure, and recreation. For every current goal or concern,

participants were asked to rate the importance on a scale from one to ten, and to indicate a time frame in which the goal/concern was expected to be accomplished or resolved. Participants were encouraged to think about goals or concerns that were relevant in the coming year. In addition to the PCI, the Behavioral Inhibition System/Behavioral Approach System scales (BIS/BAS; [52]) and the Habit Index of Negative Thinking (HINT; [53]) were used as distractor questionnaires to make the goal of our study less obvious to our participants. The PCI and BIS/BAS questionnaires were administered in Dutch (translated from English), the HINT in English (original language) given that no validated translation was available.

**Experimental session.** The SART in this experiment was based on the task used by van Vugt and Broers [2] and McVay and Kane [50]. Our SART included 720 Dutch words as stimuli that were presented in black. The majority of words were lower-case go stimuli (n = 640, 89% of total set), while only a small set were upper-case no-go stimuli (n = 80; 11% of total set). Participants were instructed to press a button as fast as they could on go stimuli, but to withhold a response on the infrequent no-go stimuli. All stimuli were presented centrally against a grey background.

Similar to the earlier works we embedded participant's *personal* current concerns in the SART task, along with the current concerns from another participant as a control (i.e., *other* concerns). We selected two personal concerns for each participant based on the PCI answers, and two 'other' concerns that were distinctly different from their personal concerns. Each current concern was translated into a triplet of words. For example, if a participant reported (A), this was translated into (B).

a. "Er zijn nog wat dingen die ik moet voorbereiden voordat ik kan beginnen met een tussenjaar."
   *"There are still some things I need to arrange before I can start taking a gap year."*

b. pauze loopbaan prepareren
   *prepare break career*

We looked for two personal concerns with the highest importance rating. Whenever two concerns had the same importance rating, we selected the most unique concern. Concerns that were too common or general were avoided. Concern words were always *go* stimuli.

The stimulus words that were not part of the personal/other concern triplets were selected from the Dutch word frequency database: SUBTLEX-NL [54]. This database contains word frequency values based on film and television subtitles. We selected the stimulus words based on the *Lg10CD* variable. This variable is a measure of the contextual diversity of a word, reflected in how many films or television shows it occurred. A validation study with a lexical decision task showed that the Lg10CD variable explained most variance in task performance (i.e., accuracy and response time; (see [54]). The same study also showed that the SUBTLEX-NL database explains 10% more variance compared to the common CELEX database (CELEX; [55]). Before selecting the word stimuli, we first discarded the least and most frequent words from the database. Thereafter, 312 words were selected with a Lg10CD value around the mean. We removed and replaced selected stimuli that were numbers, non-words, or high-arousal words.

We measured the occurrence of self-generated thought and the stickiness of thought by periodically including thought probes in the task. Thought probes consisted of two questions (Fig 2). The first question was adopted from Unsworth and Robison [27] and addressed the current thought content or *attentional state*. This question differentiated six types of attentional state: 1) on-task focus, 2) task-related interference (TRI), 3) concern related thought, 4) external distraction, 5) mind wandering, and 6) mind blanking/inattentiveness. The second

What were you just thinking about?

1) I was completely focused on the task.
2) I was evaluating aspects of the task.
   (For example my performance or how
   long it is taking)
3) I was thinking about personal matters.
4) I was distracted by my environment.
   (For example sound, temperature, my
   physical state)
5) I was daydreaming/ I was thinking of task-
   unrelated matters.
6) I was not paying attention, but neither was
   I thinking about anything specifically.

Press the number that matches your answer
to continue

How difficult was it to disengage from the
thought?

1) Very difficult
2) Difficult
3) Neither difficult nor easy
4) Easy
5) Very easy

Press the number that matches your answer
to continue

**Fig 2. Thought probe question.** An English translation of the thought probe questions used in the experiment. The first (left) question was used to measure attentional state, the second (right) question to measure stickiness of thought.

question was adopted from van Vugt and Broers [2] and asked how "sticky" the current thoughts were. Stickiness was measured as thought being 1) very sticky, 2) sticky, 3) neutral, 4) non-sticky, and 5) very non-sticky. We included 48 thought probes in the experiment.

It is relevant to note that the second 'stickiness' question has only been used once in previous research (see [2]), while the question on attentional state (and similar counterparts) have been used more frequently. Therefore, the reliability and validity of the measure cannot be guaranteed. However, there are signals that provide confidence in the reliability and validity of the stickiness question. First, the significant differences in task performance across the different levels of stickiness reported by the study of van Vugt and Broers does indicate that participants are able to report on the stickiness of their thought with similar accuracy to other thought responses. Furthermore, Mills et al. [11] showed that participants' assessment of the extent to which their thoughts were constrained, a concept similar to our stickiness, correlated significantly with external reviewers' assessments.

## Apparatus and set-up

Participants completed the PCI, HINT, and BIS/BAS questionnaires online, prior to coming to the lab, using Google Forms. The SART was performed individually in the lab. This lab contained a desk for the participants on which a computer, monitor, eye tracker, and head-mount was located. Pupil size and gaze position of the dominant eye were recorded at a sampling rate of 250 Hz using an Eyelink 1000 eye tracker from SR Research. The experiment was programmed in Psychopy (version 1.83.04; [56]) and interfaced on a Mac mini running Windows 7. The stimuli were presented on a 20 inch LCD monitor with a resolution of 1600x1200 pixels (4:3 aspect ratio) and a refresh rate of 60 Hz.

## Procedure

**Questionnaire session.**  Following registration for the experiment, participants received an email with a single link to the three online questionnaires. Participants started with the HINT, followed by the BIS/BAS and PCI questionnaire respectively. They were instructed to complete the questionnaires no later than the day before the laboratory session. After filling out the questionnaires and before the experimental session, we collected the current concerns from the answers on the PCI as described in section *Materials: Experimental Session*. The selected concerns, together with the concerns of another participant, were subsequently embedded in the stimulus set of the respective participant.

**Experimental session.**  The experimental session started with setting up the eye tracker. Participants were seated in front of the display computer and monitor, eye tracker, and head-rest. The head-rest was adjusted to the height of the participant. We performed a nine-point calibration and separate validation using the eye tracker software. The calibration and valida-tion procedure were performed for the dominant eye of the participant, or in some cases the other eye if that provided a better signal. Following calibration and validation, the instructions for the experiment were presented on the screen. The instructions on how to perform the SART were presented first, including one example of a go and a no-go trail. Afterwards, partic-ipants were informed that they would be periodically asked to report on their current thoughts. The questions for attentional state and stickiness of thought were presented on the screen, including the instructions on how to report their answer. The participants were not otherwise instructed or trained on how to use the thought probes but were invited to ask questions any time. A short practice session followed the instruction phase. This practice session consisted of ten SART trials (including one no-go trial) and one thought probe. The practice session included no trials reflecting a *personal* or *other* concern. After practice, the experiment started and the eye tracker started recording.

Each trial (see Fig 3, bottom) started with an inter-trial interval (ITI) of variable duration between 1500 and 2100 ms. During the ITI, a fixation cross consisting of the '+' symbol was presented centrally on the screen. The ITI was followed by the presentation of the stimulus word for 300 ms. *Go* stimuli were presented in lower-case, whereas *no-go* stimuli were pre-sented in upper-case. Participants were instructed to only respond on go trials (as fast as possi-ble) by pressing the 'm' key on the keyboard and to withhold a response on no-go trials. After stimulus presentation, a mask ('XXXXXXX') was presented for 300 ms followed by a response interval of 3000 ms marked by a '+' symbol. Pupil responses were recorded during the stimulus, mask, and response intervals. Once a participant responded during the mask or response interval, the experiment immediately moved on to the next ITI.

The 720 trials in the experiment (640 go; 80 no-go) and 48 thought probes were equally dis-tributed across eight blocks of 90 trials (80 go; 10 no-go) and six thought probes. All partici-pants saw the same (no concern) stimulus words but in a random order. The blocks consisted of two similar sequences of 45 stimulus words and three thought probes. The only difference between the two sequences in a block was the concern condition. One sequence contained a *personal* concern triplet, whereas the other contained an o*ther* concern triplet. The order was counterbalanced across the experiment. Furthermore, each block contained only one of the two *personal* and *other* concerns. Which type of concerns was selected alternated between blocks. When a concern triplet was presented, the order of the trial type (i.e., *go*–personal con-cern, *go*–other concern, *go*–no concern, *no-go*) and thought probes was fixed. This order was based on the experiment of McVay and Kane [50]. As shown in Fig 3 (top), concern triplets were always followed by four go (no concern) trials, one no-go trial, and one thought-probe. The thought probe questions always immediately followed the no-go trial to ensure that the

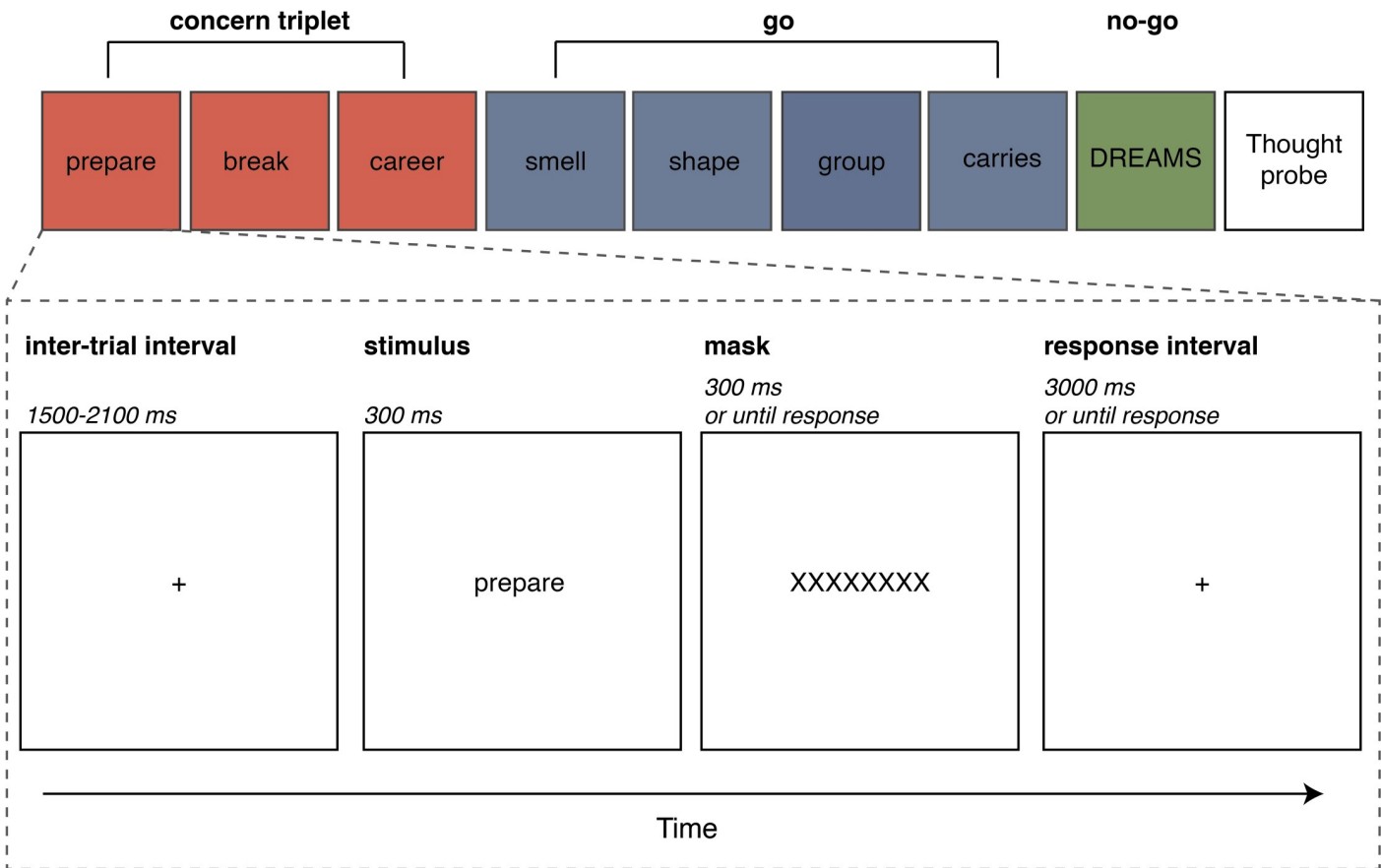

**Fig 3. Task overview.** A series of trials (top) and a single trial (bottom). Whenever a concern triplet was presented (red boxes), this was followed by four go trials (blue boxes), one no-go trial (green box), and one thought probe. Each trial, go or no-go, started with a variable inter-trial interval (ITI). Thereafter, the stimulus was presented in lowercase for go trials and uppercase for no-go trials. The stimulus was followed by a mask (until response) and a response interval (until response). Whenever the participant responded during the mask or the response interval, the experiment immediately proceeded with the next trial (i.e., ITI is drawn).

reported thought content and its stickiness could be reliably attributed to the trials before it. We are aware that a limitation of this design is that participants may confabulate their answer to the thought probe as being off-task when an error has been made on the no-go trial. Nevertheless, since this is the procedure used many prior studies on which we based our work, we kept this design.

## Data analysis

**Preprocessing of eye tracking recordings.** Before analysis, we first removed pupil size measurements associated with blinks and other artifacts. Blinks were detected using the eye tracker software. We removed the pupil size measurements marked as a blink including 100 ms before and after the event. In addition, we removed sudden upward or downward jumps. Jumps were identified by first z-scoring the pupil size timeseries for each participant individually. Subsequently, we marked pupil size measurements that had a 0.05 absolute difference in pupil size from the previous measurement (i.e., 4 ms earlier with a 250 Hz sample rate) including 20 ms before and after the observation. Subsequently, we visually inspected the marked segments of the data that would be removed with this cut-off. We concluded that this cut-off was sensitive enough to remove the jumps, but not so sensitive that it would also discard

'normal' increases in pupil dilation. In total we discarded 12.2% of the pupil size measurements with SART trials, with percentages ranging from 1.7% to 29.4% across individual participants. Trials with more than 25% discarded/missing data were removed completely, resulting in the removal of 11.1% of the trials (range = 0.3% - 57.3%). We downsampled the data to 50 Hz, taking the median pupil size for each time bin. We did not interpolate the data, since our analysis methods (generalized additive mixed models and linear-mixed effect models; see *Statistical Analysis*) can deal with missing data. After downsampling, we segmented the pupil size measurements in timeseries for individual trials ranging from 500 ms before stimulus onset to 2000 ms after onset. This time window was chosen to fully capture the pupil response to the task stimuli, while preventing overlap in the segments of neighboring trials.

**Eye tracking measures.**   We calculated the baseline pupil size by first taking the mean of the pupil size measurements in the window of 500 ms before stimulus onset. Subsequently, the baseline pupil size was then determined by z-scoring the means for each participant individually. Z-scoring the baseline values sets the grand average for each participant at zero, thereby removing individual differences in pupil size. Task evoked pupil size was obtained by subtracting the (non-transformed) baseline of each trial from the pupil size measurements in that trial.

**Behavioral measures.**   We measured task performance in the SART on accuracy, response time, and variability in response time (RTCV). Accuracy was expressed as a binomial dependent variable, coding correct responses as '1' (i.e., button press on go trials and no response on no-go trials) and incorrect responses as '0' (i.e., no response on go trials and a button press on no-go trials). Response was measured in milliseconds, but log-transformed to account for right-skewness in the distribution of these measurements. RTCV was calculated by taking the mean of the four go trials preceding a no-go trial (Fig 4), divided by the standard deviation of

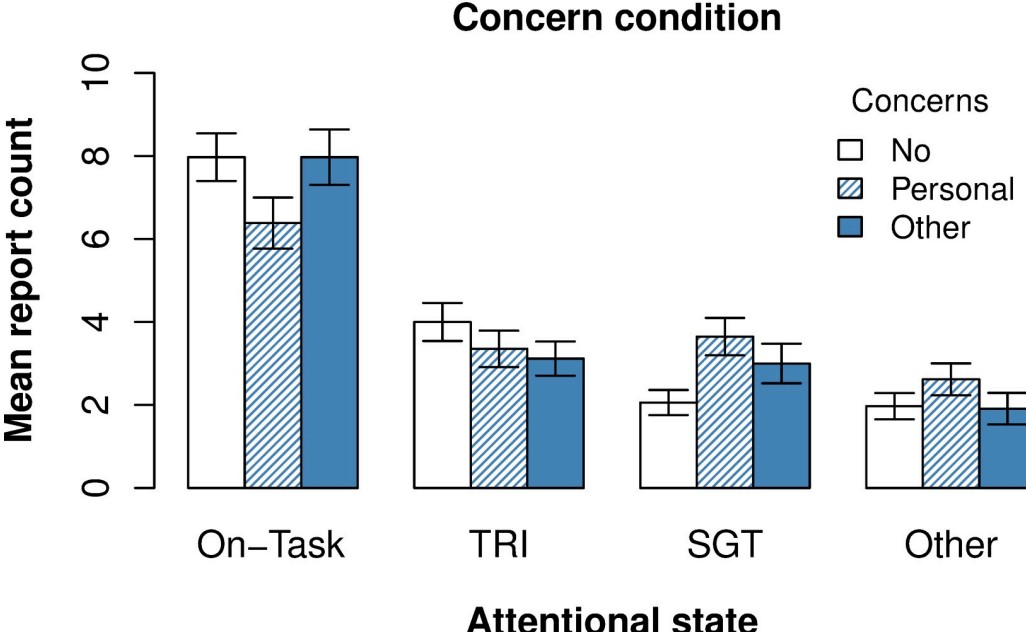

**Fig 4. Influence of concern manipulation on frequency of different attentional states.** Report counts were derived from the first thought probe question. This question had six answer options. From left to right on the x-axis, *on-task* refers to the answer option for task focus, *TRI* to task-related interference, *SGT* (i.e., *self-generated thought*) to answers on mind wandering and thoughts on personal concerns, and *other* to answers on external distraction and inalertness. Error bars reflect one standard error of the subject mean.

**Table 1. Distribution of responses to attentional state question.** Average number of responses (out of $N = 48$) to each answer option on the attentional state question per subject. Relative frequencies, expressed in percentages, are presented in the third column.

| Answer option | Frequency (out of $N = 48$ per subject) | Percentage |
|---|---|---|
| On-task | 22.3 | 46.5 |
| Task-related interference | 10.5 | 21.8 |
| Current concerns | 4.9 | 10.1 |
| External distraction | 5.2 | 10.8 |
| Mind wandering | 3.9 | 8.0 |
| Inalertness | 1.3 | 2.3 |

response time in those trials. Similar to the response times, the RTCV values were log-transformed prior to analysis.

Alongside these task performance measures, we included variables for the current concerns condition, attentional state (i.e., on-task, self-generated thought, task-related interference etc.), and stickiness level (i.e., very non-sticky, non-sticky, neutral, sticky, very sticky) in the analysis. Stickiness level was both used as (ordered) categorical dependent variable and predictor depending on the analysis. Current concerns condition and attentional state were only included as categorical predictors. The current concern condition predictor indicated whether a go/no-go trial was preceded by a triplet of personal or other concerns, or in cases where there were no concern related trials, within a window of eight trials (see Fig 4). The attentional state and stickiness level predictor indicated the answer on the first and second thought-probe question respectively.

We noticed that some answer options on both thought-probe questions had very few observations (see Tables 1 and 2). To increase the amount of observations per answer option and thereby increase statistical power, we decided to combine the answer options into larger categories. For attentional state, we combined the answer option for thoughts about current concerns (option 3) with mind wandering (option 5) into the larger category of *self-generated thought*, justified by the idea that thoughts about concerns are a special case of mind wandering. We also combined the option for external distraction (option 4) and inalertness (option 6) into the *other* category. This resulted in the following levels: on-task, task-related interference, self-generated thought, and other. For stickiness level, we decided to group the first two answer options into a *sticky* category, and the last two into *non-sticky*. We refer to the third (intermediate) answer option as *neutral*.

**Statistical analysis.** We investigated the thought probe reports by computing a count for each answer option to both questions for each participant. The resulting answer frequencies were analyzed using generalized linear models assuming a Poisson distribution. We used linear-mixed effects modeling (LME) in the remaining analyses, except when 'time' (i.e., time-in-

**Table 2. Distribution of responses to stickiness question.** Average number of responses (out of $N = 48$) to each answer option on the stickiness question per subject. Relative frequencies, expressed in percentages, are presented in the third column.

| Answer option | Frequency (out of $N = 48$ per subject) | Percentage |
|---|---|---|
| Very sticky | 3.4 | 7.4 |
| Sticky | 12.4 | 25.9 |
| Neutral | 19.8 | 41.1 |
| Non-sticky | 7.9 | 16.5 |
| Very non-sticky | 4.5 | 9.4 |

trial, time-on-task) was considered as a predictor. We assumed a Gaussian distribution for fitting response time, RTCV, and baseline pupil size. A binomial distribution was assumed for accuracy measurements. We fitted an ordered categorical LME for predicting stickiness level.

When fitting LMEs, it is important to determine a good random effects structure. Including too few random effects makes the model potentially over-confident, resulting in more Type I errors [57]. Including too many random effects lowers statistical power [58,59]. To balance Type 1 error and statistical power, we determined the random effects structure of each model using a chi-square log-likelihood-based backwards model fitting procedure. With this procedure we removed one term from the random effects structure at every step, starting from the most complex model. We kept the simpler model if the more complex model did not significantly explain more variance. Random effects that correlated strongly ($r > 0.5$) with one or more other random effects were always removed. Models that did not converge or provided a singular fit were not considered. In such cases, we continued the procedure of leaving one term out at every step. We considered trial number, block number, and participant number as random intercepts. Current concern condition, attentional state, and stickiness level were considered to have random slopes whenever they were included as a fixed effect in the model.

Statistical significance of individual predictors in the fitted LMEs were determined using chi-square log-likelihood ratio tests, testing the model including the predictor against an intercept-only model. Interactions were tested by comparing a model with the interaction against a model with only the main effects. Predictors in the LMEs were categorical. Consequently, the test statistics only reflect comparisons to a reference group of the categorical predictor(s). The reference group for attentional state was 'on-task', for stickiness of thought 'neutral', and for current concerns the 'no concern' condition. Regression estimates (i.e., intercept and slopes) of individual LMEs were transformed back to the original scale to enhance interpretation. For Gaussian LMEs we did not determine p-values, but we use report t-statistics to indicate statistical significance ($|t| \geq 2$).

We conducted timeseries analysis (e.g., for task-evoked pupil response; time-on-task effects on attentional state, stickiness, and baseline pupil size) using a nonlinear regression technique called generalized additive mixed modeling (GAMM; [60,61]). Unlike existing related research using summarizing measures such as mean pupil size after stimulus onset [40], the slope of pupil size [39,40,62], and the maximum pupil size in a specified time window [33], GAMM allows you to model full time courses. The difference from linear regression is that the slope estimates are replaced by *smooth functions* that describe how a timeseries measure such as task-evoked pupil size changes over time. When a categorical predictor is added to the GAMM, the model will fit a different smooth function for every level of this predictor. Such smooth functions can subsequently be visualized to examine the development of the statistical effects over time. GAMM also allows for including nonlinear random effects called *random smooths*. In essence, random smooths estimate random effects coefficients for the intercept as well as how the slope of a timeseries changes over time. In our analyses, we used a random smooth for *events* that reflected the individual time course of each trial and participant. Alongside a random smooth for events, we also included a nonlinear interaction between the x and y gaze position in each GAMM. This was to account for influences of gaze position on pupil size [63,64]. An issue with modeling task-evoked responses in pupil size is that the residuals of the model are not normally distributed. To account for non-normality, we fitted all GAMMs for task-evoked pupil responses (except for one) assuming a scaled-t distribution [65]. Only the GAMM estimating the influence of stickiness on go-trial evoked responses was fitted as a Gaussian model, since the model did not converge when assuming a scaled-t distribution. Another common issue is that the pupil size recordings are highly correlated over time, violating the method's assumption that residuals are independent. Violation of this assumption may

cause a GAMM model to underestimate the size of standard errors. We accounted for autocorrelation by including an autoregressive AR(1) error model within each GAMM [66]. For an excellent tutorial paper on how to use GAMMs for pupil size analysis, we refer to van Rij and colleagues [67].

From the fitted GAMMs, we could determine whether estimated smooth terms were statistically significant. In other words, we could determine whether there were significant (nonlinear) changes in the value of a dependent variable (such as task-evoked pupil size) along the time course of a trial for different attentional states and stickiness levels. We checked for significant differences between two timeseries by determining a difference curve based on the estimated smooth terms. Two timeseries were considered to be significantly different at some point in time when the estimated difference curve including a pointwise 95% confidence interval did not include zero (given that zero indicates the absence of a difference).

Preprocessing and data analysis were performed in R. We used the *lme4* package to fit the Gaussian and binomial LMEs (version 1.1–19; [68]). GAMMs and ordered categorical LMEs were fitted using the *mgcv* package (version 1.8–28; [69]). Model estimates and diagnostics for GAMMs were visualized with the *itsadug* package (version 2.3; [70]). The data and the analysis code for preprocessing and model fitting are available online at: https://osf.io/m6ujg/.

## Results

### Thought reports

First, we analyzed the reports collected from the two thought probe questions. We assessed whether embedding participant's personal concerns influenced participants' tendency to engage in sticky or off-task thinking. Next, we examined whether time-on-task influenced *attentional state* (answer to the question "what were you just thinking about?") and *stickiness* of thought (answer to the question "how difficult was it to disengage from the thought?"). Finally, we investigated the relationship between attentional state and the experienced stickiness of thought.

**Current concerns manipulation.** Fig 4 shows the effect of the current concerns manipulation. Following *no concern* triplets, we observed that on-task reports were most frequent ($M$ = 7.97 (in count) out of $N$ = 48 total reports; $SD$ = 3.34), followed by reports of task-related interference ($M$ = 4.00; $SD$ = 2.67), self-generated thought ($M$ = 2.06; $SD$ = 1.76), and other reports ($M$ = 1.97; $SD$ = 1.85). The number of self-generated thought reports increased after a personal concern triplet relative to a no concern triplet ($M_{diff}$ = + 1.59). The average increase in self-generated thought reports after concerns was significant ($\beta$ = + 1.59 (in count), $z$ = -2.12, $p$ < .001). In addition, we found that concerns from another participant increased the amount of self-generated thought reports ($M_{diff}$ = + 0.94; $\beta$ = + 0.94 (in count), $z$ = -2.73, $p$ = .03). However, the mean increase in frequency of self-generated thought following such "other concerns" was found to be smaller compared to personal concerns ($M_{diff}$ = - 0.65; $\beta$ = - 0.65 (in count), $z$ = -2.52, $p$ = .009). Therefore, while personal concerns and other concerns were both found to increase self-generated thinking, personal concerns were more potent.

With respect to the stickiness of thought, we found that participants most frequently reported their thought as neutral following *no concern* triplets ($M$ = 6.71 (out of 48 total reports); $SD$ = 4.48), followed by sticky ($M$ = 5.06; $SD$ = 3.85), and non-sticky ($M$ = 4.24; $SD$ = 4.36). In contrast to what we found for self-generated thought, we did not find support for an increase in stickiness of thought following personal (or other) concerns ($\chi^2(2)$ = 3.82, $p$ = .15). Therefore, it is unclear whether processing current concerns in the SART could increase the stickiness of thought.

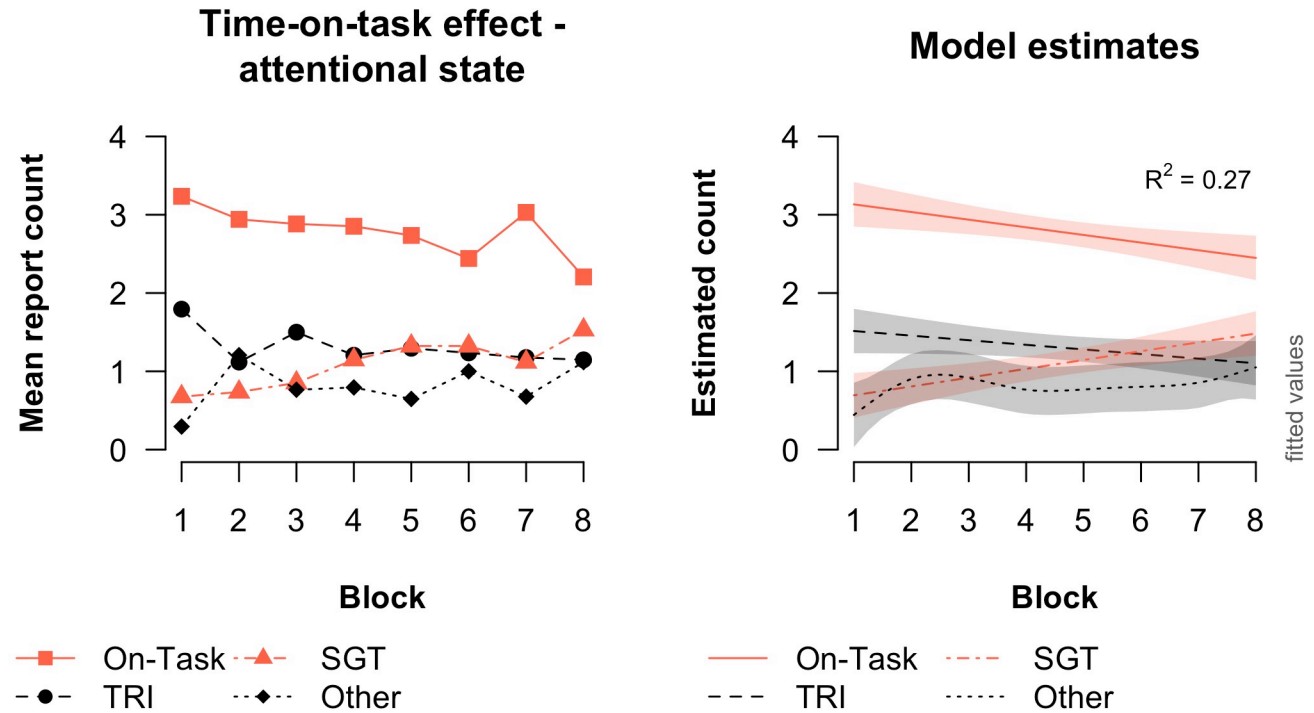

**Fig 5. Observed and estimated effect of time-on-task on attentional state.** The left plot presents the observed data and the right plot presents the estimated data from the best-fitting GAMM model. The GAMM model explains 27% of the variance in the thought probe data, calculated by taking the square of the correlation between the observed (fitted) and predicted data. Error bars in the right plot reflect estimated pointwise 95% confidence intervals.

**Time-on-task influence.** Fig 5 shows how attentional state (e.g., on-task, self-generated thought etc.) changed over the course of the experiment. The right figure shows the estimates from the fitted GAMM. Our results indicated that only the smooth terms for on-task and self-generated thought were significant (on-task: $F = 8.00$, $p = .005$; self-generated thought: $F = 10.66$; $p = .001$; task-related interference: $F = 2.90$, $p = 0.09$; other: $F = 1.33$, $p = 0.31$). Therefore, we can (only) conclude for on-task and self-generated thought that the amount of reports on this type of thinking changed over time. As shown in Fig 5, on-task thought decreased while self-generated thought increased as the task progressed.

We then asked how stickiness of thought changed over the course of the experiment. We fitted an ordered-categorical GAMM to test how time-on-task influenced the likelihood of reporting having neutral, sticky, or non-sticky thoughts. For this analysis we included the reported answer options as an ordinal dependent variable (1 being non-sticky, 2 neutral, and 3 being sticky). Block number was included as continuous predictor reflecting time-on-task. The results showed that the smooth term for block number was significant ($\chi^2 = 12.11$, $p < .001$), indicating that the reported level of stickiness changed over the course of the experiment. To inspect how the likelihood of reporting the different levels of stickiness changed over time, we obtained the predicted probability estimates from the model and plotted these in Fig 6 (right). With increasing time-on-task, we found that reports of neutral thought remained relatively constant. Furthermore, neutral thought was most prevalent in general. At the same time, we found that the probability of sticky thought increased with time-on-task, while it decreased for non-sticky thought. Together, this indicates that thought became more sticky as the task progressed.

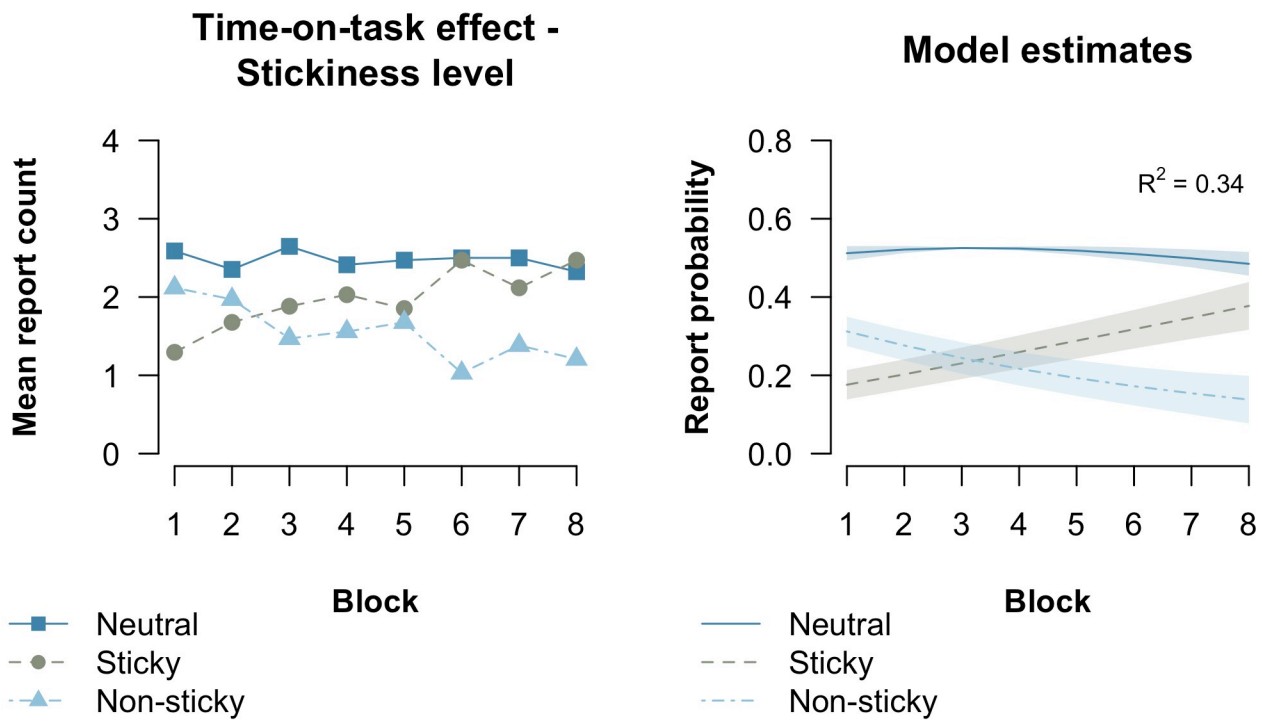

**Fig 6. Observed and estimated effect of time-on-task on stickiness level.** The left plot presents the observed data and the right the estimated data from the best-fitting GAMM model. The estimated probabilities in the right plot were derived by fitting an ordered-categorical GAMM model. Error bars in the right plot reflect estimated pointwise 95% confidence intervals.

**Relationship between attentional state and stickiness level.** We then examined whether the level of stickiness depended on whether a participant was focused on the task, mind-wandering or elsewhere. As shown in Fig 7, the reported stickiness of on-task thought strongly differed from the distracted attentional states. The majority of non-sticky ($M = 0.58$) and neutral thought reports ($M = 0.61$) were associated with on-task thought. On the other hand, reports of sticky thought were relatively more frequent in the distracted states. To test whether distracted states were experienced as stickier, we fitted an ordered categorical (ordinal) LME predicting stickiness level by attentional state. The model indicated that all off-task states were reported as stickier than on-task (on-task: intercept $\beta$ = -0.40 (transformed), $t$ = -1.79; self-generated thought: $\beta$ = + 1.53 (transformed), $t$ = 9.85; task-related interference: $\beta$ = + 1.54 (transformed), $t$ = 11.07; other: $\beta$ = + 1.77 (transformed), $t$ = 10.11).

## Task performance

We analyzed task performance to examine how attentional state and stickiness of thought was reflected in performance on go and no-go trials. Overall, we found that participants were 56.62% accurate ($SD$ = 49.57%) on no-go trials. The mean response time to go trials was 375.51 ms ($SD$ = 94.14 ms), with a mean coefficient of variance (RTCV) of 0.14 ($SD$ = 0.12). As expected, we found that all 'distracted' attentional states were associated with a lower accuracy on no-go trials compared to on-task ($\chi^2(3)$ = 216.08, $p < .001$; on-task: intercept $\beta$ = 0.80, $z$ = 5.92, $p < .001$; self-generated thought: $\beta$ = - 0.36, $z$ = -9.47, $p < .001$; task-related interference: $\beta$ = - 0., $z$ = -11.54, $p < .001$; other: $\beta$ = - 0.42, $z$ = -10.08, $p < .001$). No significant

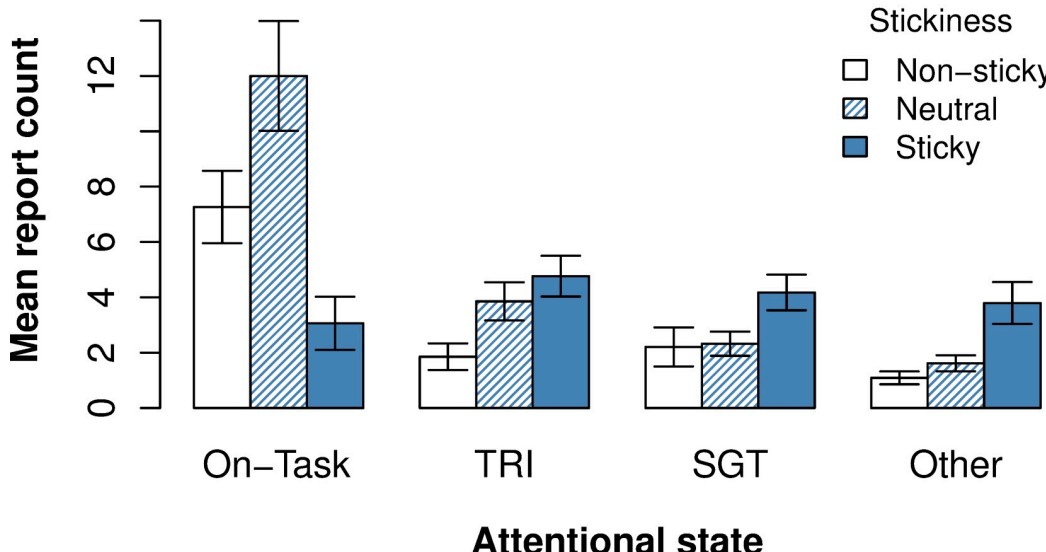

**Fig 7. Relationship between stickiness of thought and attentional state.** Report counts were derived from the first (attentional state) and second thought probe question (stickiness of current attentional state). Error bars reflect one standard error of the subject mean.

influence of attentional state was found on go response time ($\chi^2(3) = 4.88$, $p = .18$) nor on RTCV ($\chi^2(3) = 5.43$, $p = .14$). For stickiness of thought (see Fig 8), the results showed neither a significant influence of stickiness on response time ($\chi^2(2) = 2.37$, $p = .31$), nor on RTCV ($\chi^2(2) = 2.68$, $p = .26$). On the other hand, we did find a significant step-wise decrease in no-go accuracy from non-sticky thought to sticky thought. Compared to neutral stickiness, participants were 20% more accurate when current thinking was non-sticky ($\beta = + 0.20$, $z = 5.96$,

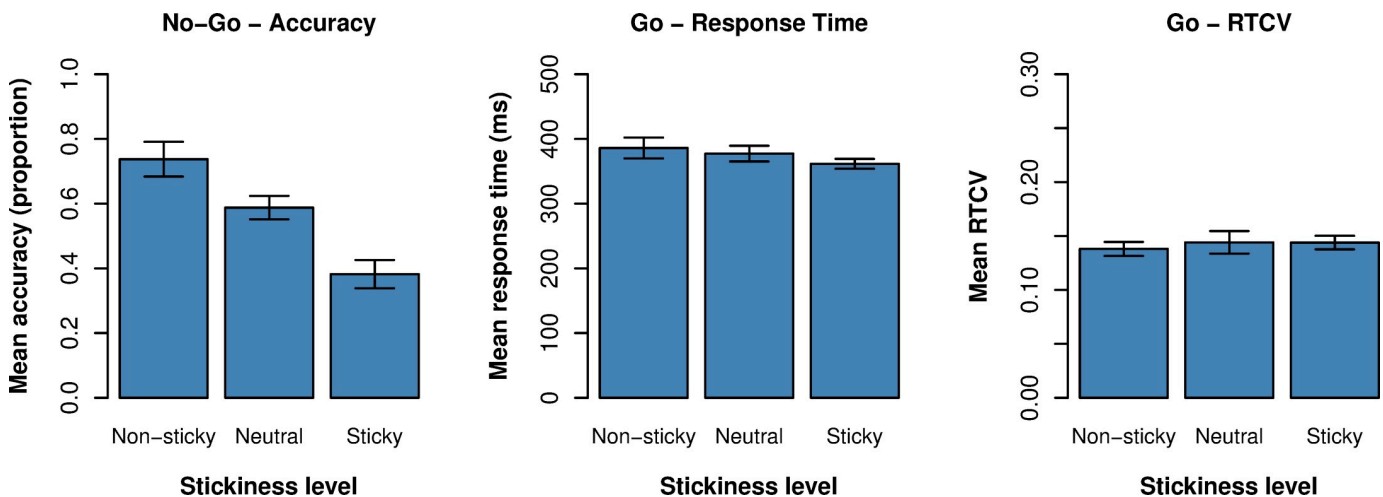

**Fig 8. Influence of stickiness on task performance.** Mean no-go accuracy (left), go response time (center), and go RTCV (right) for each level on the stickiness dimension. Error bars reflect one standard error from the subject mean.

$p < .001$), and 29% less accurate when current thinking was more sticky than neutral ($\beta = -0.29$, $z = -8.12$, $p < .001$). When attentional state was added to the LME model as an additional categorical factor alongside stickiness, we found that stickiness remained a significant predictor of no-go accuracy ($\chi^2(2) = 82.99$, $p < .001$), but not RT ($\chi^2(2) = 1.14$, $p = .57$) or RTCV ($\chi^2(2) = 0.83$, $p = .66$). This suggests that stickiness exerts unique influence on no-go accuracy on top of attentional state. The model predicted that participants were 23% more accurate when self-generated thinking was non-sticky ($\beta = + 0.23$, $z = 5.35$, $p < .001$) compared to neutral (intercept $\beta = 0.48$, $z = -0.25$, $p = .80$). Participants were 17% less accurate when self-generated thought was reported as sticky ($\beta = - 0.17$, $z = -4.63$, $p < .001$).

## Baseline pupil size

The behavioral results indicated that the frequency of different attentional states and stickiness of thought changed over the course of the experiment. Therefore, we need to take time-on-task into account when we assess how attentional state and stickiness are reflected in baseline pupil size. Fig 9 (top panel) shows the baseline pupil size across blocks for each attentional in the data (left) and predicted by a GAMM model (right). The data and the model demonstrated that baseline pupil size became smaller as the task progressed. At the same time, we failed to find consistent differences in the baseline pupil size between the attentional states. Therefore, we cannot conclude that baseline pupil size was predictive of experiencing a specific attentional state. As shown in Fig 9, bottom panel and assessed with a GAMM, we also failed to find consistent differences in baseline pupil size between the different stickiness levels.

**Task-evoked response in pupil size.**   We assessed the task-evoked response in pupil size for each attentional state and stickiness level separately for go and no-go trials. For all following analyses, we only considered correct trials. We present the grand averages of the task-evoked pupil responses along with the estimates of a fitted GAMM model in Figs 10 and 11, for go and no-go trials respectively.

**Go trials.**   What is noticeable from the task-evoked pupil responses on go trials is that there appear to be two peaks in the pupil response. The first peak occurs at around 700 ms, followed by a second peak at approximately 1200 ms. Although it is difficult to determine what is precisely reflected in these two peaks, it is reasonable to assume that the first peak reflects the amount of attention allocated to the (visual) processing of the stimulus, while the second peak may reflect processing related to the response and/or processing of the mask or fixation cross. Our results showed that the evoked response in pupil size was smaller at the first peak, but not at the second peak, when participants were engaged in self-generated thought (t = [434–788 ms]) or other distractions (t = [333–939 ms]) compared to being on-task. For task-related interference we found no significant difference in the task-evoked pupil response from on-task.

With respect to the stickiness of thought, we found that the task-evoked response in pupil size was smaller when participants experienced sticky thought compared to neutral thought (t = [283–1419 ms]), as well as non-sticky thought (t = [611–737; 965–1167 ms]). However, the task-evoked response during non-sticky thought was not found to different from the response during neutral thought.

**No-go trials.**   Similar to the go trials, we found that the evoked response in pupil size on no-go trials was characterized by two peaks occurring at around the same time points as we observed for the go trials. Self-generated thought was found to be associated with a substantially smaller response in pupil size compared to on-task for the majority of the response (t = [384–2000 ms]). For the other distracted states, we found no significant differences in pupil size compared to being on-task.

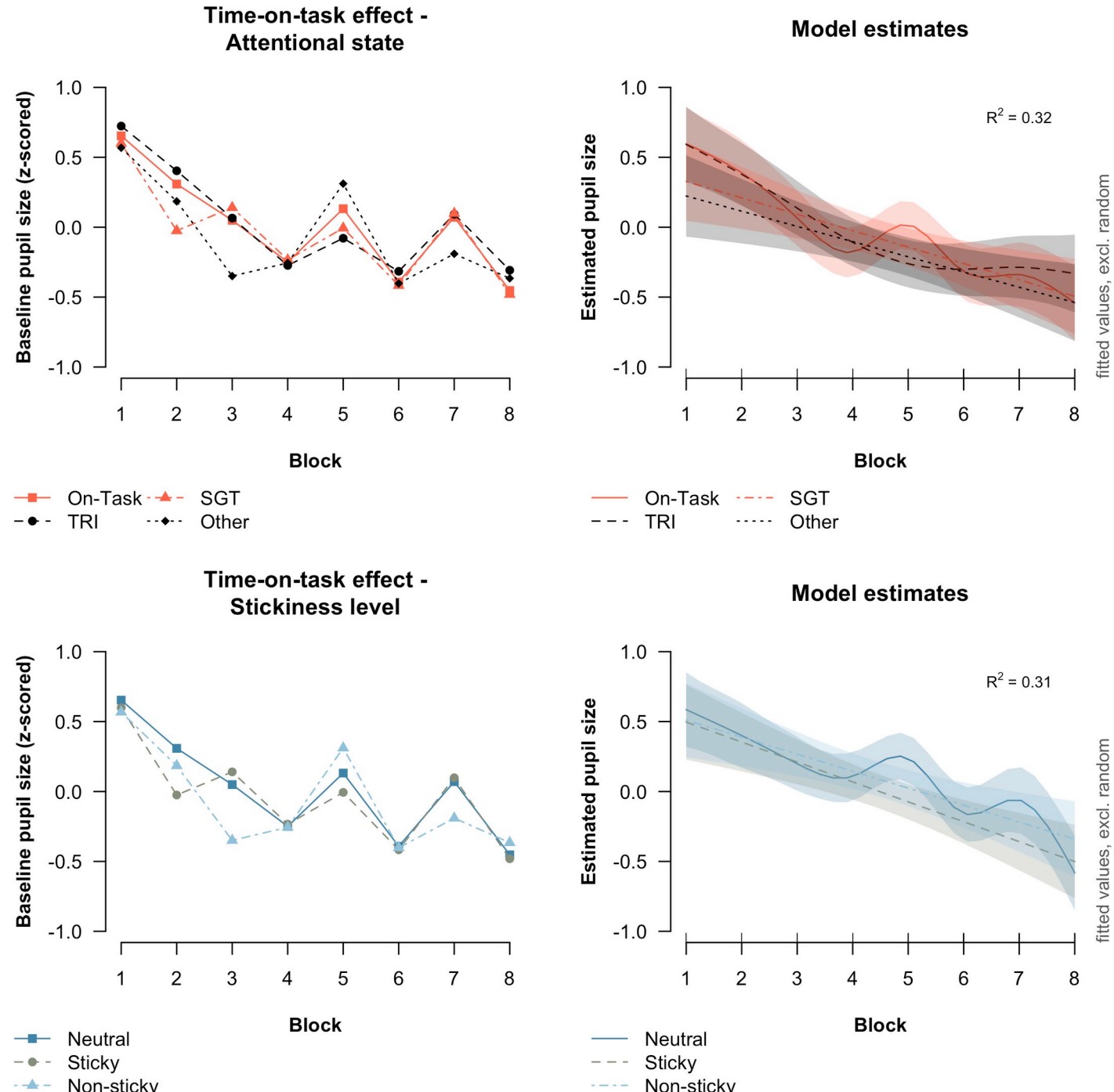

**Fig 9. Observed and estimated baseline pupil size for different attentional states and stickiness levels.** The left plots show the observed baseline pupil size across blocks for different attentional states (top panel) and stickiness levels (bottom panel). The right plots show the estimated baseline pupil size from the best-fitting GAMM. Error bars in the right plots reflect the estimated 95% confidence intervals.

When participants reported having sticky thoughts, we found that the task-evoked response in pupil size was significantly smaller compared to neutral thought (t = [510–864; 914–1672 ms]). Also for non-sticky thoughts we found that the evoked response in pupil size was smaller

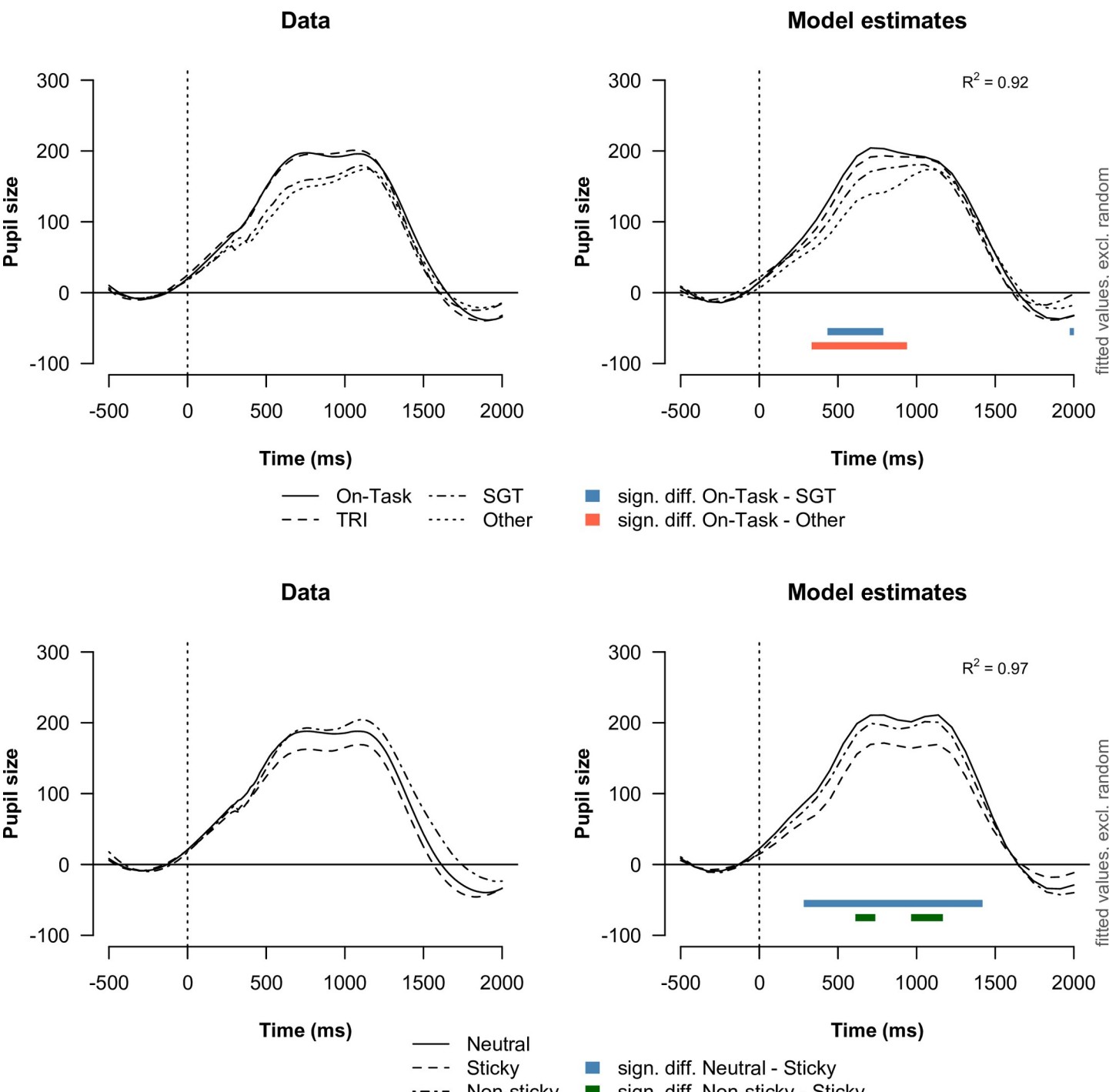

**Fig 10. Task-evoked response in pupil size aligned to go stimulus onset (t = 0).** The left plots show the average evoked response for each attentional state (top) and stickiness level (bottom) as observed in the data. The right plots show the estimates of the best-fitting GAMM models. We checked for significant differences between two evoked responses by determining a difference curve based on the estimated evoked responses. Two evoked responses were considered to be significantly different at a particular point in time when the pointwise 95% confidence interval around the estimated difference curve not include zero. We indicated a significant difference between two evoked responses with a colored bar.

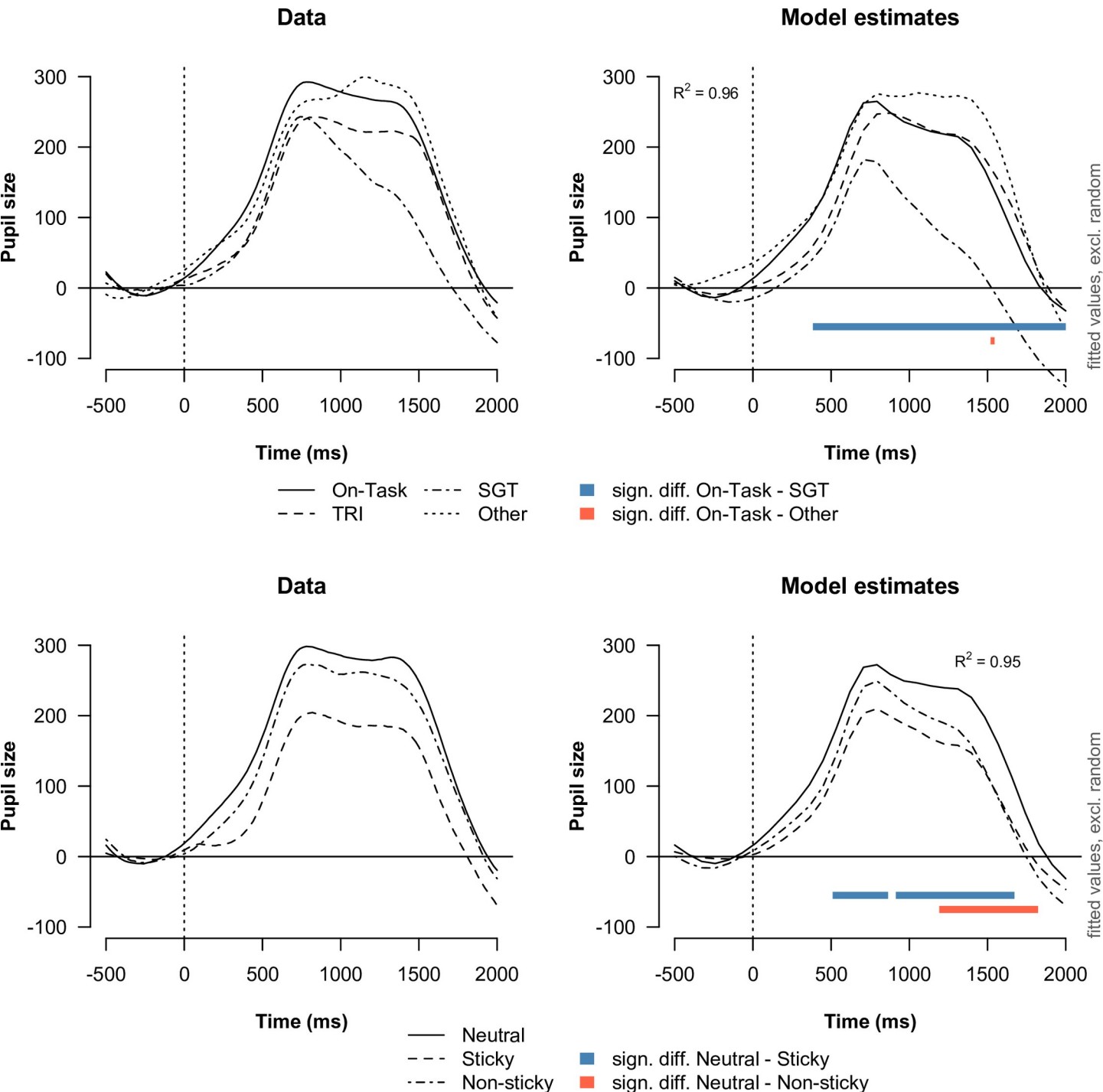

**Fig 11. Task-evoked response in pupil size aligned to no-go stimulus onset (t = 0).** The left plots show the average evoked response for each attentional state (top) and stickiness level (bottom) as observed in the data. The right plots show the estimates of the best-fitting GAMM models. Two evoked responses were considered to be significantly different at a particular point in time when the pointwise 95% confidence interval around the estimated difference curve not include zero. Significant differences between two curves were indicated with colored bars in the plot.

compared to neutral thought, but this difference only reached significance at the second peak (t = [1192–1823 ms]). The difference in evoked response between sticky and non-sticky thoughts was not found to be significant at any timepoint.

## Discussion

The goal of this research was to explore the "stickiness" dimension of ongoing thought, which reflects a participants' experienced difficulty of disengaging from thought ([2]; see also [3]). We investigated how self-reported stickiness was associated with the participant's attentional state, how it influenced task performance, and how it influenced pupil size. We adopted a variation of a sustained attention to response task (SART), which has been shown to be sensitive to lapses of attention [71–73]. Personal concerns of the participants were embedded in the SART to potentially increase the probability of observing sticky thought [16,50].

### Correlates and insights for the stickiness dimension of thought

We found that when participants reported having sticky thoughts, they also frequently reported being disengaged from the task (see also [2]). Conversely, non-sticky thought (i.e., easy to disengage) and neutral thought (i.e., neither hard nor easy to disengage) were mostly associated with being focused on the task. Therefore, the results of the present experiment demonstrated that– at least in the context of sustained attention–ongoing thought is frequently experienced as difficult to disengage from off-task thought, but easy to withdraw from task focus.

On *go* trials, we found that reports of sticky thought could be discriminated from neutral or sticky thought in task-evoked pupil dilation, but not in behavioral indices. In contrast to earlier studies (see e.g., [2,31]), this research did not demonstrate faster response times (RT) and higher variance in response times (RTCV) to go trials when participants engaged in sticky, off-task, thinking. The absence of this effect was not an issue of power. Calculating Bayes factors separately for RT and RTCV demonstrated that the present study provides strong evidence for similar RT ($BF_{01}$ = 37.3) and RTCV ($BF_{01}$ = 26.2) across different degrees of sticky thinking. An explanation for the present results may be in the relationship between RTCV and the degree to which participants were disengaged from the task (see [73]). Increases in RTCV have been associated with a state of "tuning out" (see [74]), where attention is partially allocated away the task while awareness to the general task context remains. The transient disengagement from the task during tuning out results in slowing and speeding of response times could lead to higher RTCV. In this experiment, participants were likely to be more strongly disengaged from the task during sticky thoughts–a state of "zoning out" [74]. According to Cheyne et al. [73], zoning out is associated with reactive and automatic responding to the task. It could be that the response time patterns as a result of automatic responding are not (measurably) different from responding during task focus.

While behavioral indices were similar, task-evoked responses did differ. We observed a smaller task-evoked response in pupil size for go trials during episodes of sticky thought, suggesting that less attention is allocated to task processing compared to during episodes of neutral or non-sticky thought. Hence, sticky thought can be detected by looking for signs of perceptual decoupling in task-evoked pupil dilation, even at a time when behavior does not appear to suffer.

While behavioral indices could not distinguish between sticky and non-sticky thought in *go* trials, accuracy on *no-go* trials did discriminate between different levels of stickiness. Participants demonstrated a higher no-go accuracy (i.e., more often withheld a response) when they reported having non-sticky thought compared to neutral thought, but performed severely worse when they experienced sticky thought. In addition to the performance decrement with sticky thinking, we observed that task-evoked pupil responses were smaller on *correct* no-go trials. Together with the smaller evoked response to go trials with sticky though, this provides further evidence that sticky thought limits attention and exertion of cognitive control to external task processing (even when the response ends up being correct). In contrast, we could not

discriminate non-sticky from neutral thought in task-evoked pupil dilation. Therefore, it is unclear whether cognitive processing leading to accurate performance differed when experiencing non-sticky or neutral thought, while average accuracy did differ. We argue that this indicates that participants could not reliably classify their thought as non-sticky or neutral. Instead, non-sticky reports may have been motivated by accuracy on the preceding no-go trial, explaining the better performance with non-sticky reports. Reports on sticky thought were likely not, or at least less, affected by no-go performance, since task-evoked pupil dilation was affected in correct trials.

The differences in no-go accuracy between sticky and neutral/non-sticky thought may provide some insight in how cognitive processing differs between these modes of thought. According to the literature on the SART, deliberate control is beneficial for performance on no-go trials. Deliberate control can be employed to sustain attention to the task (e.g., [75]), but also to support a controlled response strategy [76–78]. Therefore, the reason why sticky thought was associated with lower performance compared to neutral/non-sticky thought may be because this mode of thought was associated with a lower level of deliberate control.

How deliberate control influences the stickiness of thought, as well as other mechanisms outside of deliberate control, may be explained by the dynamic framework of spontaneous thought (see [10]). This framework posits that the flow of content and orientation of thought can be constrained either through cognitive control (referred to as 'deliberate constraints'), or through sensory and affective salience (referred to as 'automatic constraints'). Deliberate constraints result in a neutral/non-sticky experience, because there is volitional control on the content and orientation of thought. On the other hand, strong automatic constraints (together with weak deliberate constraints) result in high stability of thought, which may make the thoughts difficult to disengage from.

The proposed relationship between the relative contribution of automatic and deliberate constraints to the stickiness of thought is supported by existing neuroimaging studies. Individuals with depression–a disorder marked by negatively valanced sticky thought [7,8]–have shown to have greater activation of the default network when engaged in experimental tasks compared to controls [79,80]. The default network is proposed to support spontaneous thought [81–83]. The increased default network activity is, furthermore, accompanied with greater activation of (emotional) salience networks, while areas associated with cognitive control have reduced activation [84]. Therefore, these results are consistent with the idea that sticky thought is mostly constrained through salience, and less so through deliberate control.

## Implications for future studies

The present study may have practical implications for future studies. To our knowledge, this study presents the first evidence that the influence of stickiness of thought on task processing can be investigated in pupillary measures shortly prior to self-report. This opens up research opportunities for research on related modes of thinking such as perseverative cognition. Research on perseverative cognition has currently primarily used questionnaires to measure participant's general tendency to engage in rumination or worry [3,7,22], and/or to retrospectively measure whether a participant engaged (at some point during the experiment) in such thought [20]. Arguably, this method is relatively imprecise. Embedding thought probes in a task, combined with continuous pupil size measurement, could allow to investigate more precisely how rumination and worry influence cognitive processing in a task.

Triangulating between thought probe reports, behavior, and task-evoked pupil dilation demonstrated that episodes of sticky thought involve reduced attention towards the ongoing task. The reduced attention to the task may point to perceptual decoupling. While the present

experiment was not designed to investigate perceptual decoupling, follow-up research may further investigate the role of perceptual decoupling in sticky thought with concurrent measures. For example, future research may consider repeating the present study with EEG to examine whether sticky thinking has different neural correlates from non-sticky task-unrelated thought. Research on self-generated thought has indicated that episodes of off-task thinking reduce early task-evoked EEG components associated with visual processing (i.e., P1, N1; [85,86]), as well as later components such as the P3 (e.g., [14]). When used to study the influence of sticky thinking, this may help to gain understanding in to what extent perceptual decoupling modulates sensory processing (i.e., magnitude of P1, N1 components) and/or later cognitive processing (P3 component) during episodes of sticky thought.

While stickiness of thought was associated with the magnitude of task-evoked pupil dilation, it remains unclear how it affects baseline pupil size. In fact, the present study suggests that there may not be a direct relationship between the experienced stickiness and baseline pupil size at all, but rather that it is mediated by time-on-task influences. In line with other SART studies, we found that baseline pupil size declined over time ([27,72]; see also [87,88]). At the same time, the frequency of sticky thought increased when the task progressed. By consequence, one might easily arrive at the false conclusion that sticky thought is associated with a smaller baseline pupil size when time-on-task influences are not considered. As demonstrated in Fig 9, there were in fact no consistent differences in the baseline pupil size between the types of thought that was reported once time-on-task influences were considered. This mediating role of time-on-task on the relationship between baseline pupil size and the experienced thought has potential implications for existing research that has looked for correlates of different kinds of ongoing thought in baseline pupil size.

Finally, our research demonstrated that that exposure to personal concerns, indeed, increased the tendency to engage in self-generated thought. Along with personal concerns, also concerns of other participants increased the tendency for self-generated thinking. Yet, the increase in self-generated thought after exposure to personal concerns was significantly larger. This stands in contrast to previous research using a similar concern manipulation [2,50]. These studies did demonstrate an increase following processing concerns, but were not successful in finding a more potent effect of personal concerns compared to other concerns. A possible explanation is that in this study, we carefully selected the personal and other concerns to make sure that they were unique. In other words, we ensured that the other concern did not overlap with any personal concerns. In the previous reports on this task, we could not find a notion of similar practice. Hence, the discrepancy in findings may potentially be explained by the degree of overlap between the self and other concern conditions.

While exposure to personal concerns was found to locally increase self-generated thinking, it did not affect the stickiness of thought. Potentially, this may highlight that a sticky mode of thinking does not reliably result from processing personal concerns in a healthy population. Research has shown that people with a strong tendency to ruminate–a particular form of negative sticky thinking—have an attentional bias towards information that describes relevant, but negative, aspects of themselves (see e.g., [21,22]). Participants in our experiment potentially did not have a strong attentional bias towards their personal concerns. For future researchers, it may be interesting to investigate whether individuals with depression and/or individuals with high trait rumination, do engage more in sticky thought after exposure to their personal concerns.

## Conclusions

To conclude, the present study found that sticky thinking is frequently experienced when we are (temporarily) disengaged from our ongoing task. Furthermore, sticky thinking was

associated with a decrease in the ability to withhold a response on infrequent targets (no-go stimuli) and smaller responses in pupil size to task events. These results demonstrate, first of all, that individuals can report on the stickiness of their thought and that the experience can be traced in task-evoked pupil dilation. Secondly, the results indicate that attention is drawn away from the task when experiencing sticky thought. The observed attentional decoupling may be the result of reduced deliberate constraints on thought, in combination with increased automatic constraints on thought, resulting in the subjective experience of sticky thinking. Future research should investigate these claims more directly.

## Acknowledgments

We would like to thank dr. Jacolien C. van Rij for her helpful suggestions and comments on our GAMM analysis.

## Author Contributions

**Conceptualization:** Mathanja Verkaik, Marieke K. van Vugt.

**Data curation:** Stefan Huijser, Mathanja Verkaik.

**Formal analysis:** Stefan Huijser.

**Funding acquisition:** Niels A. Taatgen.

**Investigation:** Stefan Huijser, Mathanja Verkaik.

**Methodology:** Stefan Huijser, Mathanja Verkaik, Marieke K. van Vugt.

**Project administration:** Marieke K. van Vugt.

**Supervision:** Marieke K. van Vugt, Niels A. Taatgen.

**Visualization:** Stefan Huijser.

**Writing – original draft:** Stefan Huijser, Mathanja Verkaik.

**Writing – review & editing:** Stefan Huijser, Marieke K. van Vugt, Niels A. Taatgen.

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
