## [Decision Letter · Decision Letter 0]

23 Jan 2020

PONE-D-19-32012

Captivated by thought: "sticky" thinking leaves traces of perceptual decoupling in task-evoked pupil size

PLOS ONE

Dear Mr. Huijser,

Thank you for submitting your manuscript to PLOS ONE. After careful consideration, we feel that it has merit but does not fully meet PLOS ONE’s publication criteria as it currently stands. Therefore, we invite you to submit a revised version of the manuscript that addresses the points raised during the review process.

As you can see below, I have received two expert reviews. Each reviewer raises a number of important points that need to be addressed. In particular, both reviewers indicate that the interpretations and conclusions with regard to the processes at work here might not be appropriately based on the data presented (see Reviewer 1, main point 2, and Reviewer 2, main point 2). This is an important issue, as PLOS ONE specifically requires that “the data presented in the manuscript must support the conclusions drawn. Submissions will be rejected if the interpretation of results is unjustified or inappropriate, so authors should avoid overstating their conclusions. Authors may discuss possible implications for their results as long as these are clearly identified as hypotheses instead of conclusions.” It is therefore essential to address these issues in a revision.

Furthermore, both reviewers have questions with regard to the sample size, the validity of the sticky thought measure, and analytical choices concerning this measure and other measures. PLOS ONE requires that sample sizes must be large enough to produce robust results, so it is necessary to address this. It is also necessary to address the validity of the measure (also see Reviewer 1, main point 1 and below), the number of comparisons, and other analytical choices, and it is necessary to report model data for all comparisons and descriptive statistics for all variables.

Both reviewers express concerns with regard to the theoretical validity of the concept of sticky thoughts. They point out potential overlap with other types of thoughts defined elsewhere in the literature, and give excellent suggestions for further improving the theoretical clarity of your manuscript. Finally, both reviewers provide an extensive list of minor issues (clarifications, typos, missing items/legends in figures, etc.) that need to be addressed.

I would like to invite a revision that addresses these issues.

We would appreciate receiving your revised manuscript by Mar 08 2020 11:59PM. To enhance the reproducibility of your results, we recommend that if applicable you deposit your laboratory protocols in protocols.io, where a protocol can be assigned its own identifier (DOI) such that it can be cited independently in the future. For instructions see: http://journals.plos.org/plosone/s/submission-guidelines#loc-laboratory-protocols

We look forward to receiving your revised manuscript.

Kind regards,

Myrthe Faber

Academic Editor

PLOS ONE

Journal Requirements:

Reviewers' comments:

Reviewer's Responses to Questions

**Comments to the Author**

1. Is the manuscript technically sound, and do the data support the conclusions?

Reviewer #1: Partly

Reviewer #2: Partly

2. Has the statistical analysis been performed appropriately and rigorously? 

Reviewer #1: Yes

Reviewer #2: Yes

3. Have the authors made all data underlying the findings in their manuscript fully available?

Reviewer #1: Yes

Reviewer #2: Yes

4. Is the manuscript presented in an intelligible fashion and written in standard English?

Reviewer #1: Yes

Reviewer #2: Yes

5. Review Comments to the Author

Reviewer #1: The manuscript describes a study investigating the pupillometric and behavioral performance correlates of a category of thought dubbed "sticky thinking", a form of perseverative thought that it is hard to disengage from. I think that the study is interesting and there is value to study this particular type of thinking; I also liked the transparency with which the methods and analyses are described, and I think that GAMMs are very promising to describe this type of data. I also have some major and minor comments that, from my point of view, could help improve the paper.

My main comments are about the two central topics in the manuscript: sticky thought, and perceptual decoupling. Regarding sticky thought, I see the authors introducing this concept as a (somehow) new category of thought. However it is not very clear (especially in the Introduction) how it differs from other types of mind-wandering or self-generated thought, and how it relates to them. The whole mind-wandering literature is plagued with a problem of semantics: researchers in the field often use different words to refer to the same thing, or the same words for different things. I think that in this context, conceptual clarity is even more important. For how it is described in the Introduction, sticky thought seems greatly conflated with general mind-wandering: e.g., the descriptions in line 39 to 43 could easily be used for mind-wandering. Sticky thought also seems conflated with rumination, a negative form of perseverative thought common in psychopathologies like major depressive disorder or anxiety disorders. I think that, if such a new concept is to be introduced in the literature, it should be clearly defined relative to existing concepts: for example, the authors say that it is a form of rigid, and narrow-focused thought process that it is hard to disengage from, but how is it different from rumination? I imagine that the authors implicitly assume that the stickiness of thought is a category of thought that is orthogonal to the valence of the thought (so that it can also be not negative), but in this case "neutral" and "positive" sticky thought should exist. Is this the case? Is there research on this, or could the authors provide examples? As the debate on the categories of thought is an ongoing one (see Seli et al., 2018, TiCS; Christoff et al., 2018, TiCS), I think it would be helpful to better define the concept of sticky thought relative to other types of thought. Moreover, for example the evidence provided in lines 46-54, or lines 65-67, is all related to rumination/worry, but the authors’ conclusions generalise to sticky thoughts: this does not seem warranted. While the idea of a “sticky” factor of thought is interesting, as it is now, I was not 100% convinced that it exists outside of rumination-like thoughts.

My other main comment is on the concept of perceptual decoupling. This refers to a hypothesized process by the brain, aimed at insulating the mental stream of thought from external (perceptual) distractions. It is an interesting hypothesis, but it is very much open to debate if such a process exists in the brain, and evidence is still limited. One previous study (Smallwood et al., 2011, Plos One) somewhat linked this hypothesized process to a reduction in pupil sizes during on-task and off-task periods of a sustained attention task; however, this is far from proving that small pupils are an index of perceptual decoupling. While reading the manuscript, I feel that the authors imply that, finding smaller task-evoked pupil responses, automatically signals a perceptual decoupling process in action (e.g. lines 145,146, or 627-632). This seems like a form of reverse inference to me. The fact that perceptual decoupling (if it exists) might be linked to smaller pupils, does not necessarily mean that finding small pupils means that the brain is decoupling from the external environment. If this hypothesized process is really in act, we should observe other concurrent measures of reduced processing of the external environment: I don't think that the current paradigm in the study allows to do that, and additionally, there appears to be evidence for no differences in behavioral indexes of RT and RT variability between sticky and non-sticky thinking (e.g., line 623). While the stickiness factor did discriminate in no-go accuracy, this is a finding also common for mind-wandering thoughts in general, and is not intrinsically linked to perceptual decoupling. It is also not clear what the mechanism that would link smaller pupil task-evoked responses to a perceptual decoupling process would be. All in all, I do not think that the current study provides enough evidence to warrant strong affirmations such as those in lines 627-632, or lines 686-688. I don't know if the authors agree, but my suggestion would be to use a much more cautious language throughout the manuscript.

Some other comments that I have that I hope the authors will find useful:

Lines 58-84: This is true but obviously not limited to sticky thought, but is one of the main obstacles in the field of mind-wandering and of consciousness research in general. I feel like the whole paragraph could be shortened if the authors have an interest in doing so, given that thought probes are pretty much the standard way in the field to study these types of ongoing thoughts.

Lines 93-95: I have read the cited study and I couldn’t find some of the results here described (e.g. off-task thought, but not stickiness specifically, seemed to be related to task accuracy). Could the authors double-check? My apologies in advance if I missed or misread something.

Line 97: One key reference that could be added here is Seli, Cheyne & Smilek (2013), JEP:HPP.

Lines 141-142: This is slightly misleading as sticky thought is a novel concept. I do believe that there is some pupillometric research on rumination (e.g. Siegle, Steinhauser, Carter, Ramel & Thase, 2003), which could be added here, if the authors think it would be interesting for their argument.

Lines 147-149: I was confused by this sentence, as it is possible to dissociate, and measure separately, baseline to task-evoked pupil responses. Could the authors clarify?

Lines 149-151: Another study that could be discussed here and potentially in other parts of the manuscript is Konishi, Brown, Battaglini & Smallwood (2017), in which the authors find smaller baseline pupils for off-task thought, and specifically for negatively valenced and intrusive thoughts. As an author of that study I have a conflict of interest in pointing to it, but it seems to have some obvious links to the present manuscript (e.g. the concepts of intrusive and negative thoughts seems very close to those of rumination/worry).

Lines 154-158: Indeed both smaller and larger baseline pupil sizes have been found in the literature (e.g. Gilzenrat, Niewenhuis, Jepma, Cohen, 2010; Smallwood et al., 2011, 2012, Van den Brink, Murphy & Niewenhuis, 2016; Van Orden, Jung & Makeig, 2000; Konishi et al., 2017; Unsworth & Robison, 2016), and it is still not 100% clear what factors account for these differences.

Lines 163-165: This seems a little bit like a filler sentence. Maybe some concrete examples could be provided.

Line 166: a “which” seems to be missing in the phrase “in we embedded”.

Lines 170-174: To me it appears that these sentences construct a false dichotomy between sticky thoughts and general off-task thought/mind-wandering. For example, all the predictions described here can relate to off-task thought too.

Lines 182-183: How was the number of participants decided?

Lines 231-232: I think I missed how the experiment included 16 personal concern triplets, if the authors selected the 2 main concerns for each participant, and then translated each into a triplet of words. Sorry in advance if I misunderstood.

Lines 249-251: Was this second question always presented, even if for example, participants reported to be on-task or externally distracted in the first question? Were the authors still analysing such cases?

Figure 2: There’s a small typo in the third question (maters instead of matters).

Line 263: A reference could be added for PsychoPy (the most recent study is Peirce, Gray, Simpson, et al., 2019, Behavior Research Methods).

Line 304-305: It seems like the fixed order would cue participants to know in advance when a thought-probe would be presented. This could be seen as an issue, do the authors have any opinions on this?

Lines 322-324: How were these cut-offs decided?

Lines 364-365: This decision seems a bit strange in the context of pupil size and arousal, as external distraction is likely arousing, the opposite of inalertness.

Lines 367-368: What was the reason? What was the point of including a 6 point scale if it’s not used in the analyses?

Lines 481-483: Apologies in advance as I’m not sure if I’ve missed it, but was this result also taking into account the overall proportion of mind-wandering reports? If not, this result would be confounded by that, as they also increase over time.

Figures 5 and 6: The legend for the plot on the right could be a bit clearer (or is missing).

Lines 511-512: There is a serious and common problem in these designs, in which a thought-probe is presented always after a target stimulus. Following a mistake, participants might confabulate and report they were not on-task. This is a hard problem to overcome, but I think that this possibility should be discussed, given that the whole field relies on self-reports.

Lines 516-519: Again, I am not 100% sure if I missed this, but was this analysis also taking into account the underlying proportion of mind-wandering thoughts?

Lines 534-535: Did the authors analyse baseline pupil sizes across attentional states with simpler models such as LMMs? Given the amount of previous research on this, I’m just wondering if the null result depends somewhat on the GAMMs.

Lines 565-567: It seems a bit strange that the difference is smaller between sticky and non-sticky than between sticky and neutral. If these are opposite of a continuous state, one would expect the difference to be bigger between sticky and non-sticky. Do the authors have any comments on this?

Figure 10: The legend here could be clearer. Maybe it’s better to use colored lines instead of different patterns. It could also be that it wasn’t very clear because the image included in the review was lo-res.

Line 654: Minor typo, missing a “to” in “compared neutral/non-sticky”.

General: Would it be possible to have a table or a description of the best fitting models for each analysis conducted?

General: Did the authors measure and check the response times to the thought-probes? It sometimes happens that some participants will start to respond very quickly to those probes (because they want to finish the experiment faster). Such fast responses should be discarded, as it is debatable if the participants can introspect and report on their previous mental states so quickly.

Reviewer #2: Thank you for inviting me to review this paper. I thought it was generally well-written and easily understandable. The concept of sticky thought also seems important and relevant to the field of thought in general. For the most part, I think the methods, procedures, and results were described well. At the same time, I have some significant concerns that I think should be addressed before publication. Many of central issues (detailed below) deal with conceptual framing, theoretical importance, and analytical choices can potentially be addressed with a substantial revision. However, at least one of the main issues pertains to sample size (here only 34 participants were used); I believe the authors may want to consider a replication or additional data collection.

One of the main concerns I have is the limited literature covered in the Introduction. Although I think the authors have written concisely, key literature is missing. For example, an expanded discussion on the relationship between sticky thought, possibly negative valence, and arousal might help; it is only briefly mentioned now. Part of this relates to the authors choice not to make a hypothesis about the direction of sticky thought with baseline activity. This choice is completely fine, but the review of relevant concepts is still missing to make this case. There are many studies regarding rumination, etc. More directly related may be Christoff et al.’s (2016) paper which explicitly discusses concepts related to sticky thought, and some of Smallwood’s multi-dimensional experience sampling studies may also prove useful.

I also found the discussion section to be somewhat speculative without many clear theoretical implications. The authors attempt to explain some of their results with relatively shallow explanations – e.g., why they did not replicate past studies and why sticky thought was not influenced by personal concerns.

Sample size is a major concern given there is no a prior power analysis mentioned. The authors cite a previous study on sticky thought which may have been used to estimate an effect size. This field typically sees small (to medium) effect sizes, and these are not necessarily addressed by using lmer. The authors do mention computing BF in the discussion for one of the results that did not replicate, but I do not think this necessarily justifies the sample size for all the various relationships tested given the relatively low effect sizes seen for mind wandering (and other related dimensions) and performance in the literature.

One of the other main concerns is on the validity of measuring sticky thought. Figure 2 and the in text wording do not match. How were participants trained/instructed about this question? The authors also note in the discussion people may not be able to discern different levels of sticky thought using their method. While I appreciate the honesty, it seems like it could be problem with the study design/materials used since the research question was not to test a measure of sticky thought but rather the findings/interpretation depend on a reliable measure.

I was surprised to see that the authors chose to treat the sticky thoughts as categorical from the outset and also their choice to bin into three categories. There also did not appear to be a consideration for the responses in other categories as part of their models. Moreover, although the authors did not dichotomize, Seli, Beaty, et al. recently made a case to avoid binning thought dimensions because it can artificially inflate rates.

Related, this also increases the number of comparisons made when binning. In general, there were a large number of tests computed here. Were the number of tests considered in the calculating significance?

During the statistical analyses section and at the outset of the results section, the authors mention time/timeseries as an important factor to consider. Based on how this shaped their entire analytical approach, I think it might be factored into the Introduction and theoretical motivation a bit earlier. The authors also bring up their goal to assess whether they could induce sticky thought, as if this was one of their main questions. However, the paper was not framed to address this as a main question from the Introduction, so I suggest making this more explicit from the outset.

Please report all descriptive statistics for all variables.

Please also provide model data on all those constructed in a Table and mention them in the text. For example, the model comparison for sticky vs non sticky in terms of task evoked pupil size is not mentioned.

Why were go and no go trials analyzed separately?

The authors also do not mention other related metrics assessing dynamic measures of thought such as freely-moving thought. The authors may also consider making a point about the fact that sticky thought appears to be different from task unrelated thought, making it an important dimension to study.

Figure 1’s caption mentions performance but it is not in the figure.

6. PLOS authors have the option to publish the peer review history of their article (what does this mean?). If published, this will include your full peer review and any attached files.

Reviewer #1: Yes: Mahiko Konishi

Reviewer #2: No

---

## [Author Response · Author response to Decision Letter 0]

21 Jul 2020

Reviewer 1

My main comments are about the two central topics in the manuscript: sticky thought, and perceptual decoupling. 

• Regarding sticky thought, I see the authors introducing this concept as a (somehow) new category of thought. However it is not very clear (especially in the Introduction) how it differs from other types of mind-wandering or self-generated thought, and how it relates to them. The whole mind-wandering literature is plagued with a problem of semantics: researchers in the field often use different words to refer to the same thing, or the same words for different things. I think that in this context, conceptual clarity is even more important. For how it is described in the Introduction, sticky thought seems greatly conflated with general mind-wandering: e.g., the descriptions in line 39 to 43 could easily be used for mind-wandering. Sticky thought also seems conflated with rumination, a negative form of perseverative thought common in psychopathologies like major depressive disorder or anxiety disorders. I think that, if such a new concept is to be introduced in the literature, it should be clearly defined relative to existing concepts: for example, the authors say that it is a form of rigid, and narrow-focused thought process that it is hard to disengage from, but how is it different from rumination? I imagine that the authors implicitly assume that the stickiness of thought is a category of thought that is orthogonal to the valence of the thought (so that it can also be not negative), but in this case "neutral" and "positive" sticky thought should exist. Is this the case? Is there research on this, or could the authors provide examples? As the debate on the categories of thought is an ongoing one (see Seli et al., 2018, TiCS; Christoff et al., 2018, TiCS), I think it would be helpful to better define the concept of sticky thought relative to other types of thought. Moreover, for example the evidence provided in lines 46-54, or lines 65-67, is all related to rumination/worry, but the authors’ conclusions generalise to sticky thoughts: this does not seem warranted. While the idea of a “sticky” factor of thought is interesting, as it is now, I was not 100% convinced that it exists outside of rumination-like thoughts.

We completely agree that the literature is plagued by too many different kinds of thought. We have clarified the concept of “sticky thought” on p.3 (also copied below). In general, we conceive of sticky thought as being similar to rumination, but not necessarily restricted to clinical contexts. Hence, we have decided to not refer to it as “rumination” because that concept tends to be restricted to clinical contexts, and here we do not make any diagnoses nor do we work with clinical samples. 

New text: “An extreme form of such sticky thought is rumination, a rigid and narrow-focused thought process that is hard to disengage from and often negative in valence and self-related (4). In general, rumination causes individuals to be unable to concentrate and devote their attention to tasks at hand because attention is focused internally instead (2). However, in contrast to rumination, sticky thoughts could also have a positive valence, for example when we are caught up in a pleasant fantasy that we do not want to let go of, or thoughts with desire for a delicious cookie keep recurring in our minds (5,6). Another related term for sticky thought is perseverative cognition. Perseverative cognition has been associated with activation of the physiological stress system, and has been proposed to play a key role in the onset and maintenance of depression (7) and anxiety (8,9). Finally, sticky thought is closely related to the concept of constrained mind-wandering (10,11). Constrained mind-wandering is a form of mind-wandering in which thoughts cannot move freely but instead are restricted to cycling through the same narrow sets of thoughts again and again. It is different in the question that is posed to the participant—while sticky refers to the experience of the participant that it is difficult to drop the thought, constrained refers to participants’ experience of thinking the same thought again and again. “

• My other main comment is on the concept of perceptual decoupling. This refers to a hypothesized process by the brain, aimed at insulating the mental stream of thought from external (perceptual) distractions. It is an interesting hypothesis, but it is very much open to debate if such a process exists in the brain, and evidence is still limited. One previous study (Smallwood et al., 2011, Plos One) somewhat linked this hypothesized process to a reduction in pupil sizes during on-task and off-task periods of a sustained attention task; however, this is far from proving that small pupils are an index of perceptual decoupling. While reading the manuscript, I feel that the authors imply that, finding smaller task-evoked pupil responses, automatically signals a perceptual decoupling process in action (e.g. lines 145,146, or 627-632). This seems like a form of reverse inference to me. The fact that perceptual decoupling (if it exists) might be linked to smaller pupils, does not necessarily mean that finding small pupils means that the brain is decoupling from the external environment. If this hypothesized process is really in act, we should observe other concurrent measures of reduced processing of the external environment: I don't think that the current paradigm in the study allows to do that, and additionally, there appears to be evidence for no differences in behavioral indexes of RT and RT variability between sticky and non-sticky thinking (e.g., line 623). While the stickiness factor did discriminate in no-go accuracy, this is a finding also common for mind-wandering thoughts in general, and is not intrinsically linked to perceptual decoupling. It is also not clear what the mechanism that would link smaller pupil task-evoked responses to a perceptual decoupling process would be. All in all, I do not think that the current study provides enough evidence to warrant strong affirmations such as those in lines 627-632, or lines 686-688. I don't know if the authors agree, but my suggestion would be to use a much more cautious language throughout the manuscript.

We made the link between task-evoked pupil size and perceptual decoupling because the magnitude of task-evoked responses in pupil size (TERPs) have been strongly linked with allocation of attention. Hence, smaller TERPs would indicate reduced allocation of attention to external stimuli and, therefore, a decoupling of attention. However, after reading this comment of the reviewer, we agree that more cautious language is warranted. The hypothesized process of perceptual decoupling is more than just allocating less attention to surroundings, but also assumes inhibitory mechanisms to protect the internal state. This, as far as we know, cannot be measured with TERPs. We changed our wording in the referenced sentences in the Introduction and Discussion section (see p. 28, 30).

New text:

While behavioral indices were similar, task evoked responses did differ. We observed a smaller task-evoked response in pupil size for go trials during episodes of sticky thought, suggesting that less attention is allocated to task processing compared to during episodes of neutral or non-sticky thought. Hence, sticky thought can be detected by looking for signs of reduced task processing in task-evoked pupil dilation, even while behavior appears similar. (p. 28)

Triangulating between thought probe reports, behavior, and task-evoked pupil dilation demonstrated that episodes of sticky thought involve reduced attention towards the ongoing task. The reduced attention to the task may point a process called perceptual decoupling. Perceptual decoupling is a hypothesized process that functions to insulate internal thought from external (perceptual) distractions, possibly through inhibitory mechanisms (15,85,86). While the present experiment was not designed to investigate this process, follow-up research may further investigate the role of perceptual decoupling in sticky thought with concurrent measures. (p. 30)

• Lines 58-84: This is true but obviously not limited to sticky thought, but is one of the main obstacles in the field of mind-wandering and of consciousness research in general. I feel like the whole paragraph could be shortened if the authors have an interest in doing so, given that thought probes are pretty much the standard way in the field to study these types of ongoing thoughts.

We shortened the paragraph by removing lines 75 – 84 (see p. 5).

• Lines 93-95: I have read the cited study and I couldn’t find some of the results here described (e.g. off-task thought, but not stickiness specifically, seemed to be related to task accuracy). Could the authors double-check? My apologies in advance if I missed or misread something.

Thank you very much for noticing this error. Indeed, in this paper we only show relations between stickiness and response time variability, not accuracy. We have corrected this in the manuscript.

• Line 97: One key reference that could be added here is Seli, Cheyne & Smilek (2013), JEP:HPP.

We added the reference.

• Lines 141-142: This is slightly misleading as sticky thought is a novel concept. I do believe that there is some pupillometric research on rumination (e.g. Siegle, Steinhauser, Carter, Ramel & Thase, 2003), which could be added here, if the authors think it would be interesting for their argument.

In response to this comment we reformulated the referenced sentence and added the suggested article to the discussion (p. 8).

New text:

Since stickiness (i.e., the difficulty in disengaging from thought) is a novel topic, no studies have directly investigated how it is reflected in baseline and task-evoked pupil size. Nonetheless, predictions can be made based on related research and the adaptive gain curve. Given the disruptiveness of sticky thought to ongoing activities, we may expect that sticky thought, similar to self-generated thinking, is associated with smaller task-evoked responses in pupil size. As predicted by adaptive gain (see Fig 1 above), a smaller task-evoked response in pupil size with episodes of sticky thought would imply that the thought process is associated with either smaller or larger than average baseline pupil size. However, which one is open to debate. In clinical samples, Siegle et al. (45) found that rumination was associated with larger baseline pupil sizes. The researchers hypothesized that this larger baseline pupil size reflected sustained emotional processing (46). In contrast, Konishi et al. (47) found that in non-clinical samples, negative and intrusive thoughts were associated with smaller baseline pupil size (48). (p. 8)

• Lines 147-149: I was confused by this sentence, as it is possible to dissociate, and measure separately, baseline to task-evoked pupil responses. Could the authors clarify?

It is indeed true that both can be measured separately, however, they are not completely unrelated. As explained in the aforementioned paragraph, the relationship between baseline (tonic) and task-evoked pupil size (phasic) is suggested to follow an adaptive gain curve. To clarify this, we added ‘As predicted by adaptive gain’ in front of the referenced sentence (p. 7).

• Lines 149-151: Another study that could be discussed here and potentially in other parts of the manuscript is Konishi, Brown, Battaglini & Smallwood (2017), in which the authors find smaller baseline pupils for off-task thought, and specifically for negatively valenced and intrusive thoughts. As an author of that study I have a conflict of interest in pointing to it, but it seems to have some obvious links to the present manuscript (e.g. the concepts of intrusive and negative thoughts seems very close to those of rumination/worry).

We thank you for the suggested article and have included a reference to it in the discussion (p. 7-8).

• Lines 154-158: Indeed both smaller and larger baseline pupil sizes have been found in the literature (e.g. Gilzenrat, Niewenhuis, Jepma, Cohen, 2010; Smallwood et al., 2011, 2012, Van den Brink, Murphy & Niewenhuis, 2016; Van Orden, Jung & Makeig, 2000; Konishi et al., 2017; Unsworth & Robison, 2016), and it is still not 100% clear what factors account for these differences.

We agree.

• Lines 163-165: This seems a little bit like a filler sentence. Maybe some concrete examples could be provided

We decided to remove sentence.

• Line 166: a “which” seems to be missing in the phrase “in we embedded”.

We added “which” prior to the “in we embedded”.

• Lines 170-174: To me it appears that these sentences construct a false dichotomy between sticky thoughts and general off-task thought/mind-wandering. For example, all the predictions described here can relate to off-task thought too.

This may indeed appear so, however, we don’t believe there is a false dichotomy here. The main reason for that is that stickiness is a dimension that deals with the dynamics of thought, whereas the on-task/off-task dimension deals with attentional state/thought content. The predictions are similar, because sticky thoughts are likely also off-task thoughts. However, that does not necessarily imply that the dimensions are not two different things.

• Lines 182-183: How was the number of participants decided?

No method was used, but we aimed for at least 30 to have sufficient numbers for computation of reliable means and to reliable estimate the random effects.

• Lines 231-232: I think I missed how the experiment included 16 personal concern triplets, if the authors selected the 2 main concerns for each participant, and then translated each into a triplet of words. Sorry in advance if I misunderstood.

There were 16 concern triplets because each of the eight blocks contained two concern triplets. We understand the confusion, because the number of blocks and triplets per block have not been discussed yet in that section. We decided to remove the concerning sentence.

• Lines 249-251: Was this second question always presented, even if for example, participants reported to be on-task or externally distracted in the first question? Were the authors still analysing such cases?

The question was always presented, irrespective of the reported thought content. All reports were analyzed. Although reporting the stickiness of thought on on-task focus and external distraction may be less trivial for a participant, we think it is still possible to judge to how sticky/difficult it is to disengage from the focus on the task. Analyzing mind-wandering reports separately would have been an interesting analysis, but the low amount of observations within and between participants precludes that.

• Figure 2: There’s a small typo in the third question (maters instead of matters).

Thank you. We corrected the typo.

• Line 263: A reference could be added for PsychoPy (the most recent study is Peirce, Gray, Simpson, et al., 2019, Behavior Research Methods).

Good suggestion. We added the reference to the text.

• Line 304-305: It seems like the fixed order would cue participants to know in advance when a thought-probe would be presented. This could be seen as an issue, do the authors have any opinions on this?

We think it is unlikely that participants could have predicted the thought probes. Although thought probes always followed five trials after a concern triplet, there were also two thought probes that were randomly inserted in the stimulus sequences. The position of the concern triplet, and consequently, thought probes in the stimulus sequences differed for each block. Furthermore, the stimulus words were different and the inserted concerns alternated between blocks. Because of all these differences between blocks, we do not think that participants recognized the pattern. 

• Lines 322-324: How were these cut-offs decided?

We first the cut-off to a reasonable number, which was 0.05 in this case. Subsequently, we visually inspected the marked the segments of the data that would be removed with this cut-off. We concluded that this cut-off was sensitive enough to remove the jumps, but not so sensitive that it would also discard ‘normal’ increases in pupil dilation.

We added this explanation to the text in the manuscript (p. 16)

• Lines 364-365: This decision seems a bit strange in the context of pupil size and arousal, as external distraction is likely arousing, the opposite of inalertness.

True, the predicted arousal for these attentional states is different. However, these states were not so relevant for our research question, and moreover, comprised such a small subset of events that individual analysis was not feasible. Hence, we grouped them in ‘Other’. 

• Lines 367-368: What was the reason? What was the point of including a 6 point scale if it’s not used in the analyses?

We decided to use the six-point scale, because this scale had already been used by Unsworth and Robison (2016) to study the relationship between attentional states and pupil size. Using the same scale would allow us to compare our results to theirs. 

• Lines 481-483: Apologies in advance as I’m not sure if I’ve missed it, but was this result also taking into account the overall proportion of mind-wandering reports? If not, this result would be confounded by that, as they also increase over time.

No, we did not. However, you are right that these results can be explained by the differences in the proportions of thought content. That said, we did not take the proportion of mind wandering into account there, because at that point in the analysis, it was our aim to investigate whether sticky thinking in general became more prevalent. Later in the paragraph ‘Relationship between attentional state and stickiness level’ and Fig. 7 we discussed the relationship between stickiness and thought content. 

• Figures 5 and 6: The legend for the plot on the right could be a bit clearer (or is missing).

The legend on the left also applies to the plot on the right. However, we understand that this may not be obvious. We added a separate legend for the right plots in Fig 5 and 6.

• Lines 511-512: There is a serious and common problem in these designs, in which a thought-probe is presented always after a target stimulus. Following a mistake, participants might confabulate and report they were not on-task. This is a hard problem to overcome, but I think that this possibility should be discussed, given that the whole field relies on self-reports.

We agree. Thank you for reminding us to mention it in the manuscript. We discussed the possibility of confabulated responses in the materials section of the Methods section (p. 15).

New text:

As shown in Fig 3 (top), concern triplets were always followed by four go (no concern) trials, one no-go trial, and one thought-probe. The thought probe questions always immediately followed the no-go trial to ensure that the reported thought content and its stickiness could be reliably attributed to the trials before it. We are aware that a limitation of this design is that participants may confabulate their answer to the thought probe as being off-task when an error has been made on the no-go trial. Nevertheless, since this is the procedure used many prior studies on which we based our work, we kept this design.

• Lines 516-519: Again, I am not 100% sure if I missed this, but was this analysis also taking into account the underlying proportion of mind-wandering thoughts?

We did not. Similar to our analysis described on line 481-483 (previous comment), we did not intent to. However, after reading your comment we agree with you that it does make a lot of sense to do it here. Therefore, we re-analyzed the influence of stickiness on task performance (ACC,RT,RTCV) by fitted a LME model with both stickiness and attentional state as predictor (reference level Neutral and SGT). We added the following text to the manuscript on p. 24:

“When attentional state was added to the LME model as an additional categorical factor alongside stickiness, we found that stickiness remained a significant predictor of no-go accuracy (�2(2) = 82.99, p < .001), but not RT (�2(2) = 1.14, p = .57) or RTCV (�2(2) = 0.83, p = .66). This suggests that stickiness exerts unique influence on no-go accuracy on top of attentional state. The model predicted that participants were 23% more accurate when self-generated thinking was non-sticky (� = + 0.23, z = 5.35, p < .001) compared to neutral (intercept � = 0.48, z = -0.25, p = .80). Participants were 17% less accurate when self-generated thought was reported as sticky (� = - 0.17, z = -4.63, p < .001).” 

• Lines 534-535: Did the authors analyse baseline pupil sizes across attentional states with simpler models such as LMMs? Given the amount of previous research on this, I’m just wondering if the null result depends somewhat on the GAMMs.

Yes, we did and the conclusions for the LMMs and GAMMs are the same. Specifically, we fitted LMMs that did control for time-on-task effects and models that did not. The LMMs that did not account for time-on-task effects demonstrated a significantly smaller baseline pupil size for self-generated thoughts and sticky thoughts. However, when accounting for time-on-task effects these results disappeared. Model comparisons with likelihood ratio tests were in favor of the LMMs with time-on-task included. Since time-on-task turned out to be relevant, we decided to report the results of GAMMs in the manuscript. GAMMs do not assume that the relationship between baseline pupil size and time-on-task effects is linear. 

• Lines 565-567: It seems a bit strange that the difference is smaller between sticky and non-sticky than between sticky and neutral. If these are opposite of a continuous state, one would expect the difference to be bigger between sticky and non-sticky. Do the authors have any comments on this?

As mentioned in the Discussion, we think that participants could not reliably classify their thought as either non-sticky or neutral. Instead, the classification was made based on the accuracy of the no-go trial prior to the thought probe. 

• Figure 10: The legend here could be clearer. Maybe it’s better to use colored lines instead of different patterns. It could also be that it wasn’t very clear because the image included in the review was lo-res.

The plots were saved at 300 dpi, so we do not expect this to be an issue for the final article. 

• Line 654: Minor typo, missing a “to” in “compared neutral/non-sticky”.

 Thank you. We corrected the typo.

• General: Would it be possible to have a table or a description of the best fitting models for each analysis conducted?

We decided not to have the tables or descriptions in the manuscript, however, all the requested information can be found in the R markdown file on OSF. In addition, we uploaded a html version of the markdown file, which allows the reader to inspect all the code used for the analysis + the output without the need of running the code. Link: https://osf.io/m6ujg/

• General: Did the authors measure and check the response times to the thought-probes? It sometimes happens that some participants will start to respond very quickly to those probes (because they want to finish the experiment faster). Such fast responses should be discarded, as it is debatable if the participants can introspect and report on their previous mental states so quickly.

Yes, we did. However, we did not check it prior to the analysis. Our thanks for pointing this out. In response to this comment, we checked the response times to the first (thought content) and second question (stickiness rating). We found that 55 responses were shorter than 1 second (M = 636 ms, min = 385 ms) accounting for 3% of all responses (N = 1631). Notable is that almost all of the responses were ‘on-task’ and ‘neutral’ (53 out of 55). We decided not the remove these observations for two reasons. 1) We expect that participants can quickly report whether they were focused on the task. 2) It is only a very small amount of observations; hence, it is unlikely that it would influence the results.

Reviewer 2

• One of the main concerns I have is the limited literature covered in the Introduction. Although I think the authors have written concisely, key literature is missing. For example, an expanded discussion on the relationship between sticky thought, possibly negative valence, and arousal might help; it is only briefly mentioned now. Part of this relates to the authors choice not to make a hypothesis about the direction of sticky thought with baseline activity. This choice is completely fine, but the review of relevant concepts is still missing to make this case. There are many studies regarding rumination, etc. More directly related may be Christoff et al.’s (2016) paper which explicitly discusses concepts related to sticky thought, and some of Smallwood’s multi-dimensional experience sampling studies may also prove useful.

We reviewed more literature in the Introduction addressing the similarities and differences between the concept of sticky thought and rumination, constrained mind wandering etc (p. 3-5). In addition, we expanded the discussion on the relationship between sticky thought and pupil size by reviewing a study on rumination and a study on intrusive and negative thought (p.7-8). 

• I also found the discussion section to be somewhat speculative without many clear theoretical implications. The authors attempt to explain some of their results with relatively shallow explanations – e.g., why they did not replicate past studies and why sticky thought was not influenced by personal concerns.

In response to this comment, we expanded the discussion on the absence of an effect on RTCV/RT (p.28) and why stickiness was not influenced by personal concerns (p. 32-33).

New text:

(p. 28): The absence of this effect was not an issue of power. Calculating Bayes factors separately for RT and RTCV demonstrated that the present study provides strong evidence for similar RT (BF01 = 37.3) and RTCV (BF01 = 26.2) across different degrees of sticky thinking. An explanation for the present results may be in the relationship between RTCV and the degree to which participants were disengaged from the task (see (73)). Increases in RTCV have been associated with a state of “tuning out” (see (74)), where attention is partially allocated away the task while awareness to the general task context remains. The transient disengagement from the task during tuning out results in slowing and speeding of response times could lead to higher RTCV. In this experiment, participants were likely to be more strongly disengaged from the task during sticky thoughts – a state of “zoning out” (74). According to Cheyne et al. (73), zoning out is associated with reactive and automatic responding to the task. It could be that the response time patterns as a result of automatic responding are not (measurably) different from responding during task focus. 

(p. 32-33): Potentially, this may highlight that a sticky mode of thinking does not reliably result from processing personal concerns in a healthy population. Research has shown that people with a strong tendency to ruminate – a particular form of negative sticky thinking - have an attentional bias towards information that describes relevant, but negative, aspects of themselves (see e.g., (21,22)). Participants in our experiment potentially did not have a strong attentional bias towards their personal concerns. For future researchers, it may be interesting to investigate whether individuals with depression and/or individuals with high trait rumination, do engage more in sticky thought after exposure to their personal concerns.

• Sample size is a major concern given there is no a prior power analysis mentioned. The authors cite a previous study on sticky thought which may have been used to estimate an effect size. This field typically sees small (to medium) effect sizes, and these are not necessarily addressed by using lmer. The authors do mention computing BF in the discussion for one of the results that did not replicate, but I do not think this necessarily justifies the sample size for all the various relationships tested given the relatively low effect sizes seen for mind wandering (and other related dimensions) and performance in the literature.

We agree that small sample sizes can be issue when effect sizes are small (such as here), however, we do think that our results are reliable. One of the key reasons we decided to use GAMMs was not necessarily the possibility to model time series, but also because these models have been used frequently to model small effects. For example, the technique is popularly used in linguistics (e.g., Lõo, van Rij, Järvikivi & Baayen, 2016; Vogelzang, Hendriks, & van Rijn, 2016; Wieling, 2018), a field in which effect sizes are usually very small. To ensure that GAMMs did not only detect small effects but at the same time was not overfitting, we used models with complex random effects and an autoregressive model to combat autocorrelation (as suggested by van Rij, 2019). This makes the models very conservative, giving us confidence that the results are reliable and replicable. 

Lõo, K., van Rij, J., Järvikivi, J., & Baayen, R. H. (2016). Individual differences in pupil dilation during naming task. In A. Papafragou, D. Grodner, D. Mirman, & J. Trueswell (Eds.), Proceedings of the 38th Annual Conference of the Cognitive Science Society (pp. 550–555). Austin, TX: Cognitive Science Society.

van Rij, J., Hendriks, P., van Rijn, H., Baayen, R. H., & Wood, S. N. (2019). Analyzing the time course of pupillometric data. Trends in hearing, 23, 2331216519832483.

Vogelzang, M., Hendriks, P., & van Rijn, H. (2016). Pupillary responses reflect ambiguity resolution in pronoun processing. Language, Cognition and Neuroscience, 31(7), 876–885. doi: 10.1080/23273798.2016.1155718

Wieling, M. (2018). Analyzing dynamic phonetic data using generalized additive mixed modeling: A tutorial focusing on articulatory differences between l1 and l2 speakers of English. Journal of Phonetics, 70, 86–116. doi: 10.1016/ j.wocn.2018.03.002

• One of the other main concerns is on the validity of measuring sticky thought. Figure 2 and the in text wording do not match. How were participants trained/instructed about this question? The authors also note in the discussion people may not be able to discern different levels of sticky thought using their method. While I appreciate the honesty, it seems like it could be problem with the study design/materials used since the research question was not to test a measure of sticky thought but rather the findings/interpretation depend on a reliable measure.

We think that participants are able to report on the stickiness of their thoughts with comparable accuracy to other thought probe responses. If this weren’t true, we would not be able to find significant relationships between task performance and sticky thought responses. We also rely on Christoff et al (2018), who showed that participants’ assessment of the extent to which their thoughts were constrained, a concept similar to our stickiness, correlated significantly with external reviewers’ assessments.

• I was surprised to see that the authors chose to treat the sticky thoughts as categorical from the outset and also their choice to bin into three categories. There also did not appear to be a consideration for the responses in other categories as part of their models. Moreover, although the authors did not dichotomize, Seli, Beaty, et al. recently made a case to avoid binning thought dimensions because it can artificially inflate rates.

We think it is questionable whether people can reliably rate the stickiness of their thought on a scale. Therefore, measuring stickiness on a few categories is a good alternative. From the outset, we did not plan to bin the categories (for stickiness or for thought content). However, since some levels contained few observations, we needed to bin some categories to obtain reliable estimates from the GAMMs analysis. 

• Related, this also increases the number of comparisons made when binning. In general, there were a large number of tests computed here. Were the number of tests considered in the calculating significance?

No, however, the GAMM and LME’s were fitted with maximized random effects structures (supported by the data). This minimizes Type 1 error. 

• During the statistical analyses section and at the outset of the results section, the authors mention time/timeseries as an important factor to consider. Based on how this shaped their entire analytical approach, I think it might be factored into the Introduction and theoretical motivation a bit earlier. The authors also bring up their goal to assess whether they could induce sticky thought, as if this was one of their main questions. However, the paper was not framed to address this as a main question from the Introduction, so I suggest making this more explicit from the outset.

In response to this comment, we added more information about the importance of time-on-task influences in the Introduction by discussing the results of Unsworth and Robison (2016) (see p. 5).

Next text at p. 5: Thought probes are short self-report questionnaires that are embedded in a task to measure the content and dynamics of current thought (25,26). They have the advantage that experiences can be caught close to when they arise. Furthermore, they allow for repeated measures of experienced thought making it possible to investigate changes in thought content over the course of the experiment. For example, Unsworth and Robison (27) used thought probes to investigate how different attentional states, such as mind-wandering and external distraction, correlated with task performance and pupil size measures in sustained attention task. The researchers observed that task performance decreased and pupil size became smaller with time-on-task. Also, they found that reports of mind-wandering were more frequent when the experiment progressed. This demonstrates that time-on-task influences are important to consider when studying self-generated thinking. 

Increasing the tendency to engage in sticky thought with the concern manipulation was not a goal of this research. However, it was of course the reason why we used the manipulation. To prevent confusion about this, we changed the wording of a specific line in the Results section (p. 20) to

“We first assessed whether embedding participant’s personal concerns influenced the tendency to engage in sticky, off-task, thinking.”

• Please report all descriptive statistics for all variables.

 We added descriptive statistics (mean and standard deviation) for thought report frequencies (p. 20-21) and task performance (p.23) to the manuscript.

• Please also provide model data on all those constructed in a Table and mention them in the text. For example, the model comparison for sticky vs non sticky in terms of task evoked pupil size is not mentioned.

To ensure the readability of the Results sections, we decide against providing the model data in the text. However, we do provide a markdown document of the Results section on OSF. There, all the model data and output can be found along with the analysis code. 

• Why were go and no go trials analyzed separately?

For two reasons. First, it is common to analyze the go and no-go trials separately in SART tasks. Therefore, doing it here as well would make it easier to compare the results with other studies. Secondly, go and no-go trials (may) require different cognitive processes to respond correctly. Specifically, no-go trials require inhibiting a habitual response.

• The authors also do not mention other related metrics assessing dynamic measures of thought such as freely-moving thought. The authors may also consider making a point about the fact that sticky thought appears to be different from task unrelated thought, making it an important dimension to study.

On p. 5, we added a more elaborate discussion on what sticky thought is and how it relates to related dynamic measures of thought (such as constrained mind-wandering). 

• Figure 1’s caption mentions performance but it is not in the figure.

We removed the concerning sentences from the description.

---

## [Decision Letter · Decision Letter 1]

8 Oct 2020

PONE-D-19-32012R1

Captivated by thought: "sticky" thinking leaves traces of perceptual decoupling in task-evoked pupil size

PLOS ONE

Dear Dr. Huijser,

Thank you for submitting your manuscript to PLOS ONE. After careful consideration, we feel that it has merit but does not fully meet PLOS ONE’s publication criteria as it currently stands. Therefore, we invite you to submit a revised version of the manuscript that addresses the points raised during the review process.

As you can see below, both reviewers positively evaluated your revised manuscript. Reviewer 2 still has a couple of important clarification questions about conceptual, methodological, and analytical details. I would like to invite a revision that addresses these points. If you address these points comprehensively in your revision, the manuscript should be acceptable for publication. Note, however, that the final decision of course depends on the quality and clarity of the invited revision.

We look forward to receiving your revised manuscript.

Kind regards,

Myrthe Faber

Academic Editor

PLOS ONE

Reviewers' comments:

Reviewer's Responses to Questions

**Comments to the Author**

1. If the authors have adequately addressed your comments raised in a previous round of review and you feel that this manuscript is now acceptable for publication, you may indicate that here to bypass the “Comments to the Author” section, enter your conflict of interest statement in the “Confidential to Editor” section, and submit your "Accept" recommendation.

Reviewer #1: All comments have been addressed

Reviewer #2: (No Response)

2. Is the manuscript technically sound, and do the data support the conclusions?

Reviewer #1: Yes

Reviewer #2: Yes

3. Has the statistical analysis been performed appropriately and rigorously? 

Reviewer #1: Yes

Reviewer #2: Yes

4. Have the authors made all data underlying the findings in their manuscript fully available?

Reviewer #1: Yes

Reviewer #2: Yes

5. Is the manuscript presented in an intelligible fashion and written in standard English?

Reviewer #1: Yes

Reviewer #2: Yes

6. Review Comments to the Author

Reviewer #1: All my previous comments have been adressed in a satisfactory manner by the authors.

Congratulations for an interesting paper!

Reviewer #2: Thank you for the opportunity to review the revised manuscript. In general, I am supportive of this paper and think the authors have made substantial progress. At the same time, there are still unresolved issues that need more attention and clarification before recommending publication.

More details are still needed on the choice to group sticky thought into three categories. 1) what were the distributions that made the authors choose to group this way; 2) can the authors confirm the model treated these as categorical rather than continuous?; 3) The way the models are reported currently makes additional aspects unclear. For example, if they were categorical, a single chi-sq/p-value may not capture the full set comparisons because R defaults to a reference group for categorical variables based on my understanding. Was the reference group sticky? More detailed model descriptions and results could help this confusion, as mentioned in my previous review.

The revision has improved the paper, but there are still some unresolved issues with terminology. I suggest the authors avoid using the term “constrained mind wandering” which may cause further confusion with respect to terminology. The papers cited for this term also do not use this term, and they may actually argue against the use of this term given their proposed definition (i.e. Christoff et al 2016). Rather, constrained “thinking” might be more appropriate given these recent, and currently unresolved debates. Moreover, the authors actually do assess what they call mind wandering in other places in a separate question, leading to further confusion.

The last sentence of the first paragraph is confusing – i.e. the one that references the differences in their methods and theory of constraints.

I still think it is relevant to mention in the methods how participants were trained/instructed to use the scale. This is critical information. Perhaps the answer is none, which would still be important to know.

I also do not think including important model comparisons on OSF is entirely sufficient. For example, p values and effect sizes are missing throughout some of the analyses. Some additional models may also be critically important to the main results, such as sticky vs non-sticky thought for evoked pupil size. However, I do appreciate the inclusion of open materials.

Why were only correct trials analyzed?

The authors have added some descriptive statistics. I suggest a table with the M an SD for all key variables, which are still missing.

7. PLOS authors have the option to publish the peer review history of their article (what does this mean?). If published, this will include your full peer review and any attached files.

Reviewer #1: **Yes: **Mahiko Konishi

Reviewer #2: No

---

## [Author Response · Author response to Decision Letter 1]

21 Nov 2020

More details are still needed on the choice to group sticky thought into three categories. 1) what were the distributions that made the authors choose to group this way;

Answer:We chose to reduce the number of categories from five to three because there was only a low number of observations for the extreme categories per subject. Grouping these small categories allowed us to render our data more reliable. To make this more transparent, we added two tables to the manuscript: Table 1 reports the frequency of responses for each category of the attentional state question, and Table 2 reports the frequency of each response for the stickiness question (p. 18)

Added Tables:

Table 1. Distribution of responses to attentional state question. Average number of responses (out of N = 48) to each answer option on the attentional state question per subject (second column). Relative frequencies, expressed in percentages, are presented in the third column.

Answer option Frequency (out of N = 48 per subject) Percentage

On-task 22.3 46.5

Task-related interference 10.5 21.8

Current concerns 4.9 10.1

External distraction 5.2 10.8

Mind wandering 3.9 8.0

Inalertness 1.3 2.3

Table 2. Distribution of responses to stickiness question. Average number of responses (out of N = 48) to each answer option on the stickiness question per subject (second column). Relative frequencies, expressed in percentages, are presented in the third column. 

Answer option Frequency (out of N = 48 per subject) Percentage

Very sticky 3.4 7.4

Sticky 12.4 25.9

Neutral 19.8 41.1

Non-sticky 7.9 16.5

Very non-sticky 4.5 9.4

2) can the authors confirm the model treated these as categorical rather than continuous?;

Answer: Stickiness level was used in the LME analysis as a categorial independent variable/predictor in all but one of the analyses. Once, stickiness was treated as a continuous (dependent) variable, to assess the influence of attentional state on stickiness (see p. 34 for the latter). Inspired by your comment, we revisited this part of the analysis. Instead of a Gaussian LME, we fitted an order categorial LME. In this analysis, stickiness level is now treated as an (ordered) categorical dependent variable. The conclusions remained unaltered.

3) The way the models are reported currently makes additional aspects unclear. For example, if they were categorical, a single chi-sq/p-value may not capture the full set comparisons because R defaults to a reference group for categorical variables based on my understanding. Was the reference group sticky? More detailed model descriptions and results could help this confusion, as mentioned in my previous review.

Answer: Although we see the added value of describing each fitted model in the Results section, the amount of fitted models would make it unreadable. That is why we tried to carefully explain the entire data analysis procedure in the Methods section and include all the performed analyses in the associated R markdown notebook. 

Indeed we failed to mention yet what is reflected by the chi-sq/p-values for the analyses with categorical predictors. In response to your comment, we added a clarification on this to our explanation of the statistical analysis in the methods section (p. 19):

“ Statistical significance of individual predictors in the fitted LMEs were determined using chi-square log-likelihood ratio tests, testing the model including the predictor against an intercept-only model. Interactions were tested by comparing a model with the interaction against a model with only the main effects. Predictors in the LMEs were categorical. Consequently, the test statistics only reflect comparisons to a reference group of the categorical predictor(s). The reference group for attentional state was ‘on-task’, for stickiness of thought ‘neutral’, and for the current concerns the ‘no concern’ condition. Regression estimates (i.e., intercept and slopes) of individual LMEs were transformed back to the original scale to enhance interpretation. For Gaussian LMEs we did not determine p-values, but we report t-statistics to indicate statistical significance (|t| ≥ 2).” (p. 19)

The revision has improved the paper, but there are still some unresolved issues with terminology. I suggest the authors avoid using the term “constrained mind wandering” which may cause further confusion with respect to terminology. The papers cited for this term also do not use this term, and they may actually argue against the use of this term given their proposed definition (i.e. Christoff et al 2016). Rather, constrained “thinking” might be more appropriate given these recent, and currently unresolved debates. Moreover, the authors actually do assess what they call mind wandering in other places in a separate question, leading to further confusion.

Answer: Thank you for the critical analysis of the terminology. We read the cited papers again and agree with you that it is more appropriate to speak about constrained thinking instead of constrained mind wandering. In our manuscript, we considered stickiness as an independent dimension of thought alongside attentional state. Hence, sticky thought does not necessarily imply mind wandering, but could be any kind of thought. Similarly, Mills et al (2018) concluded from their study that freedom of movement is independent from attentional state. Claiming that sticky thought is related to constrained mind wandering creates unnecessary confusion and conflicts with the cited work.

New text: Finally, sticky thought is closely related to the concept of constrained thinking (10,11). Constrained thinking refers to an experience in which thoughts do not move freely but instead are focused on a narrow set of content. (p. 3)

The last sentence of the first paragraph is confusing – i.e. the one that references the differences in their methods and theory of constraints.

Answer: What we try to explain there is that our concept of stickiness refers to the experienced difficulty of dropping a current stream of thought, whereas constrained refers to the experience of having a stream of thought that is – deliberately or not – focused on a narrow set of content. We revised the sentence to make this clearer.

New text (p. 3): It is different from our concept of sticky thinking in the question that is posed to the participant—while sticky refers to the experience of the participant that it is difficult to drop the current stream of thought, constrained refers to participants’ experience of having a stream of thought that is – deliberately or not – restricted to a narrow set of content.

I still think it is relevant to mention in the methods how participants were trained/instructed to use the scale. This is critical information. Perhaps the answer is none, which would still be important to know.

Answer: The participants were not trained on how to use the thought probes, but they were shown the thought probe questions prior to the experiment including instructions on how to report an answer. We revised the text on p.14 to include this information.

New text: Following calibration and validation, the instructions for the experiment were presented on the screen. The instructions on how to perform the SART were presented first, including one example of a go and a no-go trail. Afterwards, participants were informed that they will be periodically asked to report their current thoughts. The questions for attentional state and stickiness of thought were presented on the screen, including the instructions on how to report their answer. The participants were not otherwise instructed or trained on how to use the thought probes. A short practice session followed the instructions, consisting of ten SART trials (including one no-go trial) and one thought probe. Participants were encouraged to ask questions when something was unclear. 

I also do not think including important model comparisons on OSF is entirely sufficient. For example, p values and effect sizes are missing throughout some of the analyses. Some additional models may also be critically important to the main results, such as sticky vs non-sticky thought for evoked pupil size. However, I do appreciate the inclusion of open materials.

Answer: P-values are indeed not reported for some of the analyses. In most cases that is because the used statistical technique does not provide them (i.e., for the Gaussian LMEs and timeseries analyses with GAMMs). How we determined statistical significance for these techniques is explained in the methods section. In other cases we did not report individual p-values (and/or regression estimates) to prevent visual clutter. For example when all contrasts were significant, or when all were not. However, to accommodate your concerns we added the individual p-values (or t-statistics for Gaussian LMEs) for these latter cases. 

New text (p. 23): Our results indicated that only the smooth terms for on-task and self-generated thought were significant (on-task: F = 8.00, p = .005; self-generated thought: F = 10.66; p = .001; task-related interference: F = 2.90, p = 0.09; other: F = 1.33, p = 0.31). Therefore, we can (only) conclude for on-task and self-generated thought that the amount of reports on this type of thinking changed over time.

New text (p. 23): We then asked how stickiness of thought changed over the course of the experiment. We fitted an ordered-categorical GAMM to test how time-on-task influenced the likelihood of reporting having neutral, sticky, or non-sticky thoughts. For this analysis we included the reported answer options as an ordinal dependent variable (1 being non-sticky, 2 neutral, and 3 being sticky). Block number was included as continuous predictor reflecting time-on-task. The results showed that the smooth term for block number was significant (�2 = 12.11, p < .001), indicating that the reported level of stickiness changed over the course of the experiment. To inspect how the likelihood of reporting the different levels of stickiness changed over time, we obtained the predicted probability estimates from the model and plotted these in Fig 6 (right).

New text (p. 24): To test whether distracted states were experienced as stickier, we fitted an ordered categorical (ordinal) LME predicting stickiness level by attentional state. The model indicated that all attentional states were reported as stickier than on-task (on-task: intercept � = -0.40 (transformed), t = -1.79; self-generated thought: � = + 1.53 (transformed), t = 9.85; task-related interference: � = + 1.54 (transformed), t = 11.07; other: � = + 1.77 (transformed), t = 10.11).

New text (p. 25): As expected, we found that all ‘distracted’ attentional states were associated with a lower accuracy on no-go trials compared to on-task (�2(3) = 216.08, p < .001; on-task: intercept � = 0.80, z = 5.92, p < .001; self-generated thought: � = - 0.36, z = -9.47, p < .001; task-related interference: � = - 0.42, z = -11.54, p < .001; other: � = - 0.42, z = -10.08, p < .001).

We also added the results for the comparison between non-sticky and sticky thoughts for evoked response size.

New text (p. 28): The difference in evoked response between sticky and non-sticky thoughts was not found to be significant at any timepoint. 

Why were only correct trials analyzed?

We only analyzed correct trials because we were interested only in investigating how being on-task or being engaged in self-generated thoughts/sticky thinking influenced task processing. Incorrect responses had a very different task-evoked response measured in pupil size. As shown in the Figure below*, there is a much higher peak in pupil size during incorrect trials versus correct trials. We are not sure what causes the higher peak, but it could be an error monitoring process. Incorrect responses were relatively infrequent, and therefore could not reliably be incorporated in the analyses. (p. 14)

*R script to generate the Figure has been added to OSF in the folder Analysis as: ‘TERP_incorrect_correct_nogo.Rmd’. Link to OSF repository: https://osf.io/m6ujg/ . 

The authors have added some descriptive statistics. I suggest a table with the M an SD for all key variables, which are still missing.

Answer: For the previous revision, we added the descriptive statistics for all key variables in the text including: frequency of different attentional states (p. 22), frequency of different stickiness levels (p. 22), no-go ACC (p. 25), no-go RT (p. 25), no-go RTCV (p. 25). Having a table alongside that is redundant.

---

## [Editor Report · Decision Letter 2]

24 Nov 2020

Captivated by thought: "sticky" thinking leaves traces of perceptual decoupling in task-evoked pupil size

PONE-D-19-32012R2

Dear Dr. Huijser,

We’re pleased to inform you that your manuscript has been judged scientifically suitable for publication and will be formally accepted for publication once it meets all outstanding technical requirements.

Kind regards,

Myrthe Faber

Academic Editor

PLOS ONE
---

## [Editor Report · Acceptance letter]

27 Nov 2020

PONE-D-19-32012R2 

Captivated by thought: “sticky” thinking leaves traces of perceptual decoupling in task-evoked pupil size 

Dear Dr. Huijser:

I'm pleased to inform you that your manuscript has been deemed suitable for publication in PLOS ONE. Congratulations! Your manuscript is now with our production department. 

Kind regards, 

on behalf of

Dr. Myrthe Faber 

Academic Editor

PLOS ONE